# Nucleoside analogue activators of cyclic AMP-independent protein kinase A of *Trypanosoma*

Sabine Bachmaier[1], Yuri Volpato Santos[1], Susanne Kramer[1,7], George Boniface Githure[1], Thomas Klöckner[1], Julia Pepperl[1], Cordula Baums[1], Robin Schenk[1], Frank Schwede [2], Hans-Gottfried Genieser[2], Jean-William Dupuy[3], Ignasi Forné[4], Axel Imhof [4], Jerôme Basquin[5], Esben Lorentzen[6] & Michael Boshart [1]

Protein kinase A (PKA), the main effector of cAMP in eukaryotes, is a paradigm for the mechanisms of ligand-dependent and allosteric regulation in signalling. Here we report the orthologous but cAMP-independent PKA of the protozoan *Trypanosoma* and identify 7-deaza-nucleosides as potent activators ($EC_{50} \geq 6.5$ nM) and high affinity ligands ($K_D \geq 8$ nM). A co-crystal structure of trypanosome PKA with 7-cyano-7-deazainosine and molecular docking show how substitution of key amino acids in both CNB domains of the regulatory subunit and its unique C-terminal αD helix account for this ligand swap between trypanosome PKA and canonical cAMP-dependent PKAs. We propose nucleoside-related endogenous activators of *Trypanosoma brucei* PKA (TbPKA). The existence of eukaryotic CNB domains not associated with binding of cyclic nucleotides suggests that orphan CNB domains in other eukaryotes may bind undiscovered signalling molecules. Phosphoproteome analysis validates 7-cyano-7-deazainosine as powerful cell-permeable inducer to explore cAMP-independent PKA signalling in medically important neglected pathogens.

[1] Biocenter, Faculty of Biology, Genetics, Ludwig-Maximilians-University Munich (LMU), 82152 Martinsried, Germany. [2] BIOLOG Life Science Institute, 28199 Bremen, Germany. [3] Centre de Génomique Fonctionnelle Bordeaux, Université Bordeaux, F-33076 Bordeaux, France. [4] Biomedical Center, Ludwig-Maximilians-University Munich (LMU), 82152 Martinsried, Germany. [5] Max Planck-Institute for Biochemistry, 82152 Martinsried, Germany. [6] Department of Molecular Biology and Genetics, Aarhus University, Aarhus 8000, Denmark. [7] Present address: Department of Cell & Developmental Biology, Biocenter, University of Würzburg, Würzburg 97074, Germany. Correspondence and requests for materials should be addressed to M.B. (email: boshart@lmu.de)

The mechanisms of protein kinase regulation, specifically by small second messenger molecules, have been studied in great detail using protein kinase A (PKA) as a paradigm[1]. PKA, discovered 50 years ago, is present in most eukaryotes except plants, has a highly conserved structure, and is the main effector of the second messenger cAMP. Hence, the synonym cAMP-dependent protein kinase is commonly used. The inactive PKA holoenzyme is a complex of regulatory (PKAR) and catalytic (PKAC) subunits, either as R-C heterodimer or $R_2$-2C hetero-tetramer. The $R_2$ homodimer is formed by an N-terminal dimerization/docking (DD) domain that also mediates sub-cellular localization via A kinase anchoring proteins (AKAPs). Two C-terminal cyclic nucleotide binding (CNB) domains cooperatively bind two molecules of cAMP, resulting in a con-formational change of the R subunit that releases the active cat-alytic kinase subunit(s) from the inhibitory pseudo-substrate or substrate site of PKAR. The CNB domain is an ancient evolu-tionarily conserved domain family with >7500 members that confers ligand-dependent allosteric regulation to a diverse range of proteins[2]. In eukaryotes, CNB domains are bound and regu-lated by cyclic nucleotides. In bacterial transcription factors, some CNB domains can bind other ligands like heme in the case of the CO sensing transcription activator CooA[3] or chlorinated phenolic compounds in CprK, a member of the ubiquitous CRP-FNR family of transcription activators[4]. In metazoans, PKA has diverse functions ranging from metabolism and gene regulation to development, motility, and memory[5]. Many of these functions are tissue-specific and compartmentalized at the subcellular level[6]. In lower eukaryotes including fungi or apicomplexan protozoa like *Plasmodium* and *Toxoplasma*, PKA plays key roles in nutrient sensing, developmental switches, or infectivity processes[7–9]. Most species encode one or two PKAR isoforms and several PKAC isoforms[10]. The resulting holoenzyme isoforms differ in cell type-specificity, developmental expression, sub-cellular localization, and affinity to cAMP, thereby accounting for the pleiotropic functions of cAMP signalling.

*Trypanosoma brucei* species are kinetoplastid parasites that infect a large variety of mammals, causing severe disease in domestic animals with important economic losses in endemic countries. The parasite is also causative of the deadly human African sleeping sickness, a neglected tropical disease[11]. Trans-mission is restricted to the habitat of the Tsetse fly in tropical Africa. Development of the parasite in the host and vector is a prerequisite for transmission. This developmental process can be induced by cAMP analogues[12–14], although this is mediated by intracellular hydrolysis products of these analogues[15] operating via a complex network of effectors[16]. The parasite has been shown to release cAMP as a mechanism of evading the host's innate immunity[17]. Essential roles of intracellular cAMP signal-ling have also been documented for cell division[12,18–20] and social motility[21]. It is therefore surprising that all attempts to detect cAMP-dependent kinase activity in African trypanosomes have failed[22–27]. Genes encoding three PKA catalytic subunit ortho-logues and one regulatory subunit orthologue have been identi-fied in the *T. brucei* genome[22,26,28], whereas alternative cAMP effectors like EPAC orthologues and cNMP-gated ion channels were not detected. By screening a genome-wide RNAi library for cAMP resistance in *T. brucei*, we identified a novel cAMP binding protein (CARP1) unique to kinetoplastids[29], yet PKA was not among the hits of the screen. The catalytic subunits of *T. brucei* PKA are highly conserved with the presence of all 11 canonical kinase subdomains, the essential threonine in the kinase activa-tion loop, and conserved residues implicated in mammalian PKAC's binding to the regulatory PKAR subunits[30]. TbPKAR has a conserved C-terminal part with two CNB domains and the PKA substrate motif (RRT̲TV) that interacts with and inhibits PKAC.

TbPKAR differs from its metazoan orthologues by an extended N-terminal domain with leucine-rich repeats (LRR) (Fig. 1a). Some amino acid substitutions of consensus residues in the cAMP binding pockets have been noticed in sequence alignments[22,31]. The link between cAMP and PKA remains elu-sive in *Trypanosoma* in spite of the excellent overall conservation of the kinase.

There is a surprising deficit of knowledge in signalling mechanisms in these phylogenetically distant organisms and no complete pathway from receptor to effector has been elucidated to date. A possible explanation for this knowledge gap is sug-gested by the domain architectures found in the kinome of try-panosomes: few known signalling domains or protein–protein interaction domains are linked to the catalytic kinase domains of the 176 identified protein kinases. In addition, unusual domain combinations prevail[32]. Many conserved signalling effectors are likely to be differently connected and wired in various pathway architectures in these phylogenetically distant protozoa. This might also be the case for some second messenger dependencies. Here, we show that *Trypanosoma* PKA is not a cyclic nucleotide-dependent protein kinase. We use a chemical biology approach to identify highly specific activators of *Trypanosoma* PKA. The first crystal structure of a kinetoplastid PKAR explains the structural requirements for ligand selectivity. We suggest that this PKA has evolved to bind novel ligand(s), possibly taking the role of second messenger(s) in *T. brucei*. Our new activators are excellent tools to study this cAMP-independent PKA signalling in trypanosomes.

## Results

**PKA holoenzymes in *Trypanosoma brucei*.** We first established that the orthologous PKA subunit genes in *T. brucei* do encode proteins able to form the expected holoenzyme complexes of regulatory (R) and catalytic (C) subunits. One allele of *PKAC1* was Ty1-epitope tagged in situ, while the second *PKAC1* allele was deleted to generate *T. brucei* cell line Δc1/Ty1-C1 (Fig. 1b). The absence of a wild type *PKAC1* allele allowed simultaneous detection of the highly similar PKAC2 isoform by a PKAC1/2-specific antibody (Fig. 1c and Supplementary Fig. 1a). PKAR was then C-terminally PTP-tagged in situ in cell line Δc1/Ty1-C1 to generate Δc1/Ty1-C1 R-PTP (Fig. 1b, c). All three PKA catalytic subunit isoforms were pulled down by PKAR-PTP from lysates of cell line Δc1/Ty1-C1 R-PTP but not from the control cell line Δc1/Ty1-C1 (Fig. 1d). Pull down from cell lines expressing Ty1- or HA-tagged PKAC or PKAR subunits independently confirmed the interactions between PKAR and each of PKAC1, 2, 3 in a heterodimeric complex (Supplementary Fig. 1b, c). No co-precipitation of untagged PKAR or other PKAC isoforms was observed with tagged PKAR or PKAC1, 2, or 3 (Fig. 1d and Supplementary Fig. 1c), indicating the absence of a tetrameric $R_2$-2C complex that is found in mammalian PKA. Heterodimeric PKAR-C complexes are not unusual in lower eukaryotes[27,33,34].

The catalytic function of the kinase is essential for growth and viability, as RNAi-mediated repression of the catalytic subunits *PKAC1/2* or *PKAC3* is lethal or growth inhibitory, respectively[35] (Supplementary Fig. 2). Severe cell division defects are detected, as cytokinesis stages and multinucleated cells accumulate. This phenotype is frequently found when targeting essential trypano-some genes[36]. It does not necessarily indicate a specific role in cytokinesis. RNAi-mediated repression of *PKAR* has a very similar phenotype due to the rapid decrease of free PKAC1 (Supplementary Fig. 2). This indicates degradation of free PKAC released upon holoenzyme dissociation, as was observed for PKAC in mammalian cells[37]. Nevertheless, cell clones can be selected that maintain a basal PKAC1 level sufficient for survival

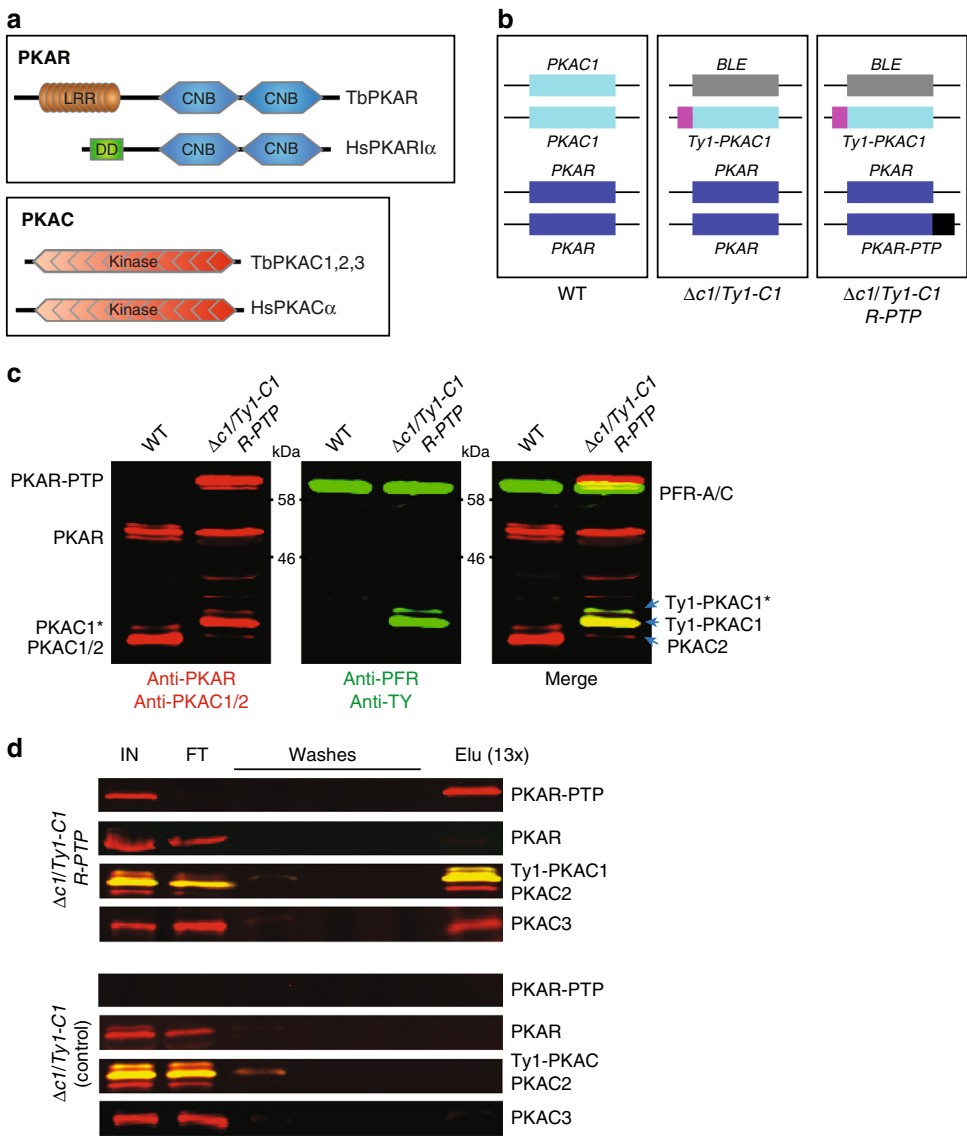

**Fig. 1** PKA holoenzyme complexes in *T. brucei*. **a** Domain architecture of PKAR (top) and PKAC (bottom) orthologues from *T. brucei* (Tb) (TriTrypDB accessions: PKAR, Tb927.11.4610; PKAC1, Tb927.9.11100; PKAC2, Tb927.9.11030; PKAC3, Tb927.10.13010) compared to human (Hs) PKA (Uniprot accessions: PKARIα, P10644; PKACα, P17612). LRR leucine-rich repeat region, DD dimerization/docking domain, CNB cyclic nucleotide binding domain, kinase kinase domain. **b** Genotypes of cell lines with in situ tagged *PKAC1* (*Ty1-C1*, cyan; Ty1 epitope tag in magenta) and *PKAR* (*R-PTP*, blue; PTP-tag in black) compared to wild type (WT). The phleomycin resistance cassette (*BLE*, grey) is indicated. **c** Two-colour fluorescent western blot analysis of the double tagged cell line (*Δc1/Ty1-C1 R-PTP*) using anti-PKAR and anti-PKAC1/2 (left panel, red signals), and anti-Ty1 and anti-PFR-A/C (loading control) (middle panel, green signals). The merge of both channels is shown on the right panel. Ty1 causes a mobility shift of PKAC1 and enables the detection of PKAC2 in cell lines devoid of wild type PKAC1. Note that PKAC1 and PKAR appear as doublet bands that we interpret as modification (PKAC1*) and allelic polymorphism, respectively, in the MiTat 1.2 cell line. **d** PTP affinity purification followed by western blot analysis of double tagged and control cell lines using antibodies as in **c** and anti-PKAC3. Equivalent amounts of soluble input material (IN), flow-through (FT), washes, and 13 equivalents of the eluate (Elu) were loaded. Source data to **c** and **d** are provided as a Source Data file

after homozygous deletion of *PKAR*. The resulting Δ*pkar*/Δ*pkar* cell lines[38] used later in this work show a mild growth phenotype with population doubling time (PDT) of 7.5 h versus 5.4 h for wild type cells.

**Trypanosoma PKA is not activated by cAMP.** The PKA holoenzymes were immunoprecipitated from trypanosomes expressing epitope-tagged PKA subunits to assay PKA activity. Contaminating kinase activities in the precipitate were excluded as (1) Ty1-affinity purification from wild type cells or cells

expressing a catalytically inactive Ty1-PKAC1 N153A mutant[39] did not pull down PKA-specific activity and (2) phosphorylation of the PKA-specific substrate kemptide was inhibited by the PKA-specific pseudo-substrate peptide PKI(5–24)[40] (Supplementary Fig. 3a). The basal activity of holoenzymes immunoprecipitated from cells expressing PKAR-Ty1 or Ty1-PKAC1 or HA-PKAC2 was not increased by cAMP (Supplementary Fig. 3b), even when the cyclic nucleotide was added at unphysiologically high concentrations (1 mM). In the same experiments, cGMP activated at ≥100 μM (Supplementary Fig. 3b). Some activation by cGMP had been noted before[22], but several lines of evidence argue against

the presence of cGMP or cGMP signalling in trypanosomatids[41], minimizing the likelihood that cGMP is a physiological activator of the kinase. The unconventional cyclic nucleotides cXMP, cIMP, cCMP, and cUMP also did not show any significant effects (Supplementary Fig. 3c). In order to exclude unsuitable assay conditions, an in vivo assay for PKA activity was established that is based on transgenic expression of the heterologous PKA substrate VASP (human platelet **va**sodilator **s**timulated **p**hosphoprotein) in *T. brucei*. Phosphorylation of VASP at the PKA site Ser-157 causes an electrophoretic mobility shift from 46 to 50 kDa[42] (Supplementary Fig. 3e). Non-phosphorylated (46 kDa) and phosphorylated VASP (50 kDa) were quantified by western blot analysis. The ratiometric determination of VASP phosphorylation (non-phosphorylated/phosphorylated) is a reliable proxy for PKA activity. The myristoylated membrane-permeable peptide inhibitor myr-PKI(14–22)[40] reduced the measured activity to background (Supplementary Fig. 3f). We then examined the effects of cAMP on PKA activity in live cells by three independent approaches. First, pharmacological elevation of intracellular cAMP was accomplished by inhibition of phosphodiesterases (PDEs) with a highly potent inhibitor of trypanosomal PDEs (CpdA, now renamed as NPD-001). The intracellular cAMP content increased up to 600-fold upon treatment with CpdA (Fig. 2a) but did not elicit any change in in vivo PKA activity (Fig. 2b). Second, reverse genetic elevation of intracellular cAMP was achieved by inducible RNAi-mediated depletion of the major cAMP-specific PDEs, *PDEB1* and *PDEB2*[20]. The resulting 45-fold increase of cAMP content (Fig. 2c) also did not stimulate PKA activity (Fig. 2d). Third, VASP expressing cells were treated with membrane-permeable cAMP analogues 8-pCPT-cAMP, 8-pCPT-2′-O-Me-cAMP, or cAMP-AM (see Supplementary Table 1). The latter is a prodrug cleaved by esterases to deliver cAMP intracellularly[43]. No PKA activation was detected for any of the analogues up to 1 mM (Supplementary Fig. 3h). Membrane-permeable cGMP derivatives also had no effect on PKA activity in vivo (Supplementary Fig. 3i).

**The drug dipyridamole induces PKA activity in trypanosomes.** The most compelling evidence against cAMP-dependent PKA activity in trypanosomes is the lack of any change in kinase activity upon an up to 600-fold increase in intracellular cAMP content, caused by the PDE inhibitor CpdA. Prior to the availability of CpdA, the same experiment was done with dipyridamole, a potent inhibitor of mammalian PDEs[44] with modest activity against trypanosomal PDEs[15,45]. Initially, we were misled by a dose- and time-dependent induction of VASP phosphorylation by dipyridamole (Fig. 2e, f, Table 1) that correlated with a moderate increase in intracellular cAMP content (Fig. 2g). As the data shown in Fig. 2a–d provide strong evidence that this effect by dipyridamole cannot be caused by the increase in cAMP, dipyridamole must induce PKA activity either directly or indirectly by a cAMP-independent mechanism. Yet, the effect is clearly mediated by PKAR, as no induction was detected in a VASP reporter cell line with homozygous deletion of *PKAR* (Fig. 2h).

**Compound screening for activators of TbPKA.** Dipyridamole, a licensed drug inhibiting thrombocyte aggregation, is a PDE inhibitor but also interferes with adenosine transport and metabolism[46]. The possible link between PKA activation and purine metabolism motivated a targeted compound screen. We tested 13 different purine nucleoside or nucleotide analogues (Supplementary Table 1) for activity in the in vivo PKA reporter assay. Most of them are predicted to be membrane permeable due to lipophilic groups. Out of these, the four 7-deazaadenosine (tubercidin, Tu) analogues toyocamycin (Toyo), 5-iodo-

tubercidin (5-I-Tu), 5-bromo-tubercidin (5-Br-Tu), and sangivamycin induced PKA activity with $EC_{50}$ values of 88 nM, 390 nM, 625 nM, and 39 μM, respectively (Fig. 3a, Table 1). The remaining compounds had either a slight inhibitory (8-pCPT-Ado, 8-pCPT-2-′O-Me-Ado) or no effect (Tu, 8-pCPT-Guo, 2-Cl-Ado, 8-pCPT-Ade, 6-Br-Tu, 8-Br-Ado, 6-Cl-PuR) in the PKA reporter assay (Fig. 3, Supplementary Fig. 4a). The most potent activator, toyocamycin, did not induce phosphorylation of VASP in the *pkar* knock out genetic background (Supplementary Fig. 4b), demonstrating that PKA is the target kinase. Growth or viability of the parasites was affected by continuous treatment with all 7-deazaadenosine analogues (Table 1). However, this was apparently due to off-target effects, as the drugs had very similar effects on *pkar* knock out cells (Table 1). Since these analogues are known to target multiple proteins and processes in mammalian cells, including kinases and synthesis of DNA, RNA, and proteins (reviewed in ref. [47]), growth or viability phenotypes were expected.

**7-Deazaadenosine analogues bind and activate TbPKAR.** The identified compounds may activate trypanosome PKA either directly or indirectly in the cell-based reporter assay. To address this, tagged PKAR and PKAC1 subunits were co-expressed in the heterologous *Leishmania tarentolae* expression system and the holoenzyme complex was isolated by tandem affinity purification (Supplementary Fig. 4e, inset). In vitro kinase assays with kemptide and [$^{32}$P]-ATP as substrates showed kinase activation upon addition of tubercidin and 5-substituted tubercidin analogues with the following order of potency: 5-I-Tu > Toyo > 5-Br-Tu > tubercidin > sangivamycin (Fig. 3a, Table 1). The $EC_{50}$ values measured with recombinant *T. brucei* holoenzyme purified from *L. tarentolae* were in the same order of magnitude as the $EC_{50}$ values of the in vivo PKA reporter assay. Up to 3.4-fold differences between in vitro kinase assay and cell-based assay are likely due to uptake, accumulation, or metabolism of the individual compounds in trypanosome cells. Tubercidin enters the trypanosome cell via nucleoside transporters[48], while the 5-substituted analogues are more lipophilic and predicted to cross the cell membrane by passive diffusion. Tubercidin uptake might be too slow to cause PKA activation within the time frame of the in vivo reporter assay. The evidence for a direct mode of action provided by in vitro kinase assays was further corroborated by measuring binding parameters between the activating compounds and purified N-terminally truncated *T. brucei* PKAR(199–499) expressed in *Escherichia coli* (Supplementary Fig. 4h). Isothermal titration calorimetry (ITC) determined a $K_D$ of 57 nM for toyocamycin with a ligand to protein molar ratio of 2.1 (±0.2), indicating availability of both CNB sites for binding (Fig. 3a, Table 1). A slightly lower $K_D$ of 32 nM was obtained for 5-I-Tu with a ligand to protein molar ratio of only 1.2 (±0.2). We suggest that 5-I-Tu has a preference for one binding site, whereas the $K_D$ value for Toyo averages over both binding sites. This is supported by the different thermodynamic signatures that indicate enthalpically-driven binding, which for 5-I-Tu is counteracted by a negative entropic effect (Fig. 3c). It should be noted that all binding assays are performed with the ligand-free (apo) form of the PKAR subunit, whereas kinase assays probe the R-C holoenzyme. Nevertheless, the binding $K_D$ values for the analogues are close to the respective $EC_{50}$ for kinase activation in vitro. Most importantly, the values for Toyo and 5-I-Tu are similar to the $K_D$ for cAMP binding to mammalian PKARIα measured under identical conditions (28 ± 5 nM, Supplementary Fig. 4f and ref. [49]). Dipyridamole, whose activation of *T. brucei* PKA had initially guided our compound screen, did not

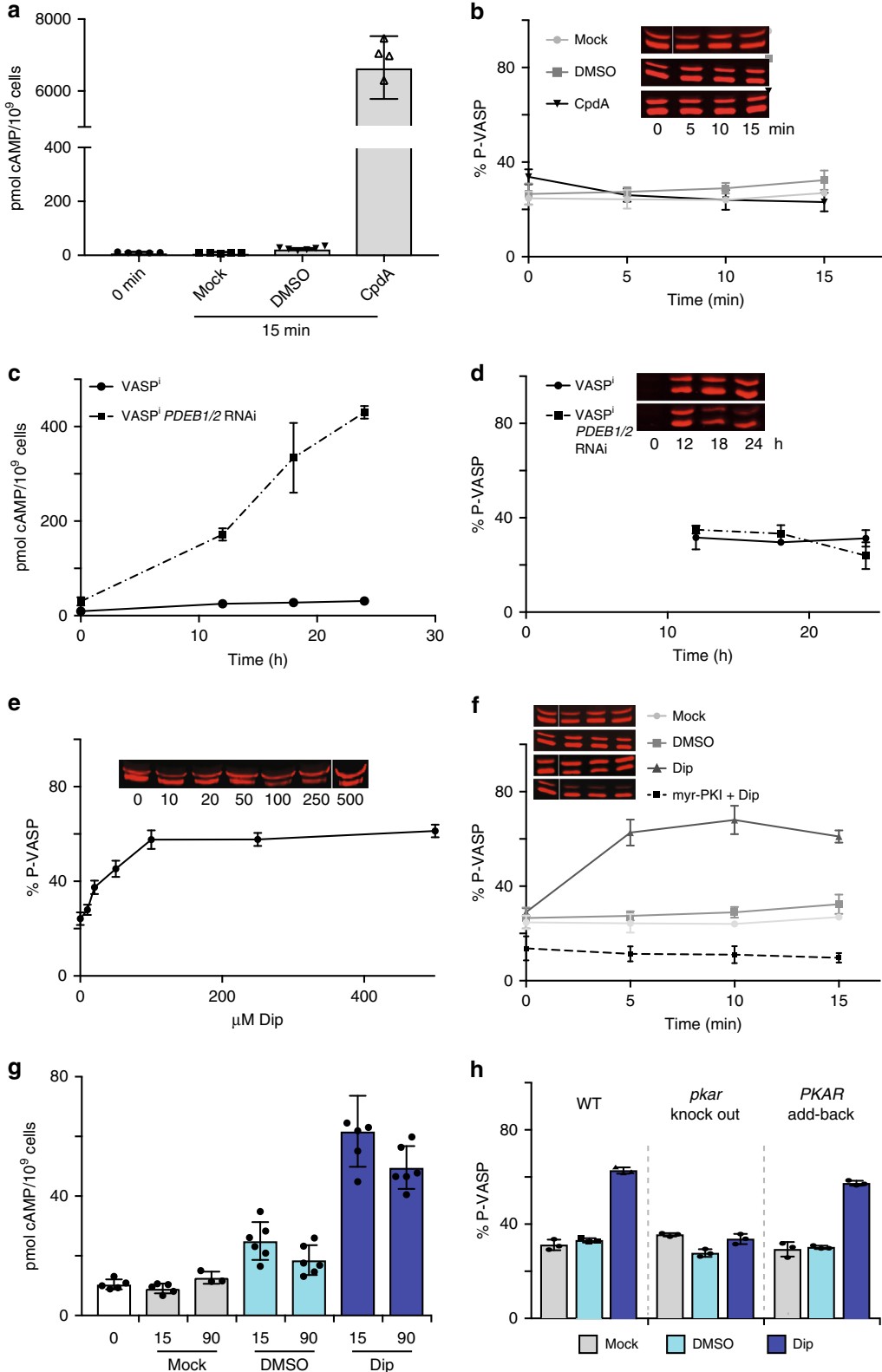

have any effect in the in vitro kinase assay (Supplementary Fig. 4e), supporting an indirect mode of action in vivo.

**Development of a trypanosome-specific PKA activator**. In mammalian cells, some nucleoside analogues have been reported to be significantly less toxic when the adenosine moiety is replaced by inosine[50,51]. Therefore, as the first step in optimization, we changed the purine ring side groups of toyocamycin (7-cyano-7-deazaadenosine) to match inosine, resulting in the compound 7-cyano-7-deazainosine (7-CN-7-C-Ino). Compared to Toyo, 7-CN-7-C-Ino had 36-fold higher activation potency on the purified holoenzyme (EC$_{50}$ 6.5 ± 4.6 nM) and 7-fold higher binding affinity to PKAR ($K_D$ 8 ± 2 nM) in ITC measurements

**Fig. 2** *T. brucei* PKA is not activated by cAMP. **a** Intracellular [cAMP] (mean ± SD of independent replicates; $n = 5$ (0 min, Mock); $n = 6$ (DMSO, CpdA)) and **b** in vivo PKA activity (with western blot inset) in cells treated or not (Mock) for 15 min with the PDE inhibitor CpdA[19] (10 µM; now renamed as NPD-001[74]) or solvent (1% DMSO). Release of cAMP into the medium upon treatment was neglectable (Supplementary Fig. 3d). **c** Intracellular [cAMP] (mean ± SD of $n = 3$ independent replicates) and **d** in vivo PKA activity (with western inset) upon inducible (1 µg ml$^{-1}$ tetracycline) RNAi repression of *PDEB1* and *PDEB2*. Kinase activity cannot be determined at time 0, since the RNAi cell line (VASP$^i$ *PDEB1/2* RNAi) and the control line (VASP$^i$) both harbour a tetracycline inducible *VASP* transgene. **e** Dose response of in vivo PKA activity upon 15 min treatment with dipyridamole (Dip). **f** Time course of in vivo PKA activity upon treatment with 100 µM dipyridamole (Dip) or dipyridamole + myr-PKI(14–22) or solvent (1% DMSO) or Mock. **g** Intracellular [cAMP] (mean ± SD of independent replicates; $n = 5$ (0 min, Mock 15 min); $n = 3$ (Mock 90 min); $n = 6$ (DMSO, Dip)) upon treatment with 100 µM dipyridamole (Dip) or 1% solvent (DMSO) or Mock for 15 or 90 min. **h** In vivo PKA activity upon treatment as in **g** for 15 min in wild type (WT), homozygous *pkar* knock out and *PKAR* add-back (in situ rescue) cells. For all in vivo kinase reporter assays, one representative western blot is shown as inset and data points are mean ± SD of $n = 3$ independent replicates. Source data are provided as a Source Data file

---

### Table 1 Summary of TbPKA activators

| Compound | EC$_{50}$ in vivo PKA assay[a] | EC$_{50}$ in vitro PKA assay[b] | $K_D$ in vitro binding to PKAR (ITC)[c] | Molar ratio of binding (ITC)[c] | IC$_{50}$ growth WT[d] | IC$_{50}$ growth *pkar* KO[d] |
|---|---|---|---|---|---|---|
| 7-CN-7-C-Ino | 838 ± 139 nM | 6.5 ± 4.6 nM | 8 ± 2 nM | 2 ± 0.2 | ≥125 µM | ≥125 µM |
| Toyo | 87.6 ± 9.6 nM | 234 ± 92 nM | 57 ± 25 nM | 2.1 ± 0.1 | 739 ± 59 nM | 690 ± 58 nM |
| 5-I-Tu | 390 ± 20 nM | 203 ± 79 nM | 32 ± 9 nM | 1.2 ± 0.2 | 330 ± 38 nM | 346 ± 41 nM |
| 5-Br-Tu | 625 ± 66 nM | 1.7 ± 1.4 µM | n.d. | n.d. | 874 ± 122 nM | 1009 ± 95 nM |
| Tubercidin | No effect | 5.0 ± 0.5 µM | n.d. | n.d. | n.d. | n.d. |
| Sangivamycin | 39.3 ± 5.6 µM | 10.8 µM | n.d. | n.d. | 713 ± 40 nM | 1548 ± 171 nM |
| Dipyridamole | 22.4 ± 4.2 µM | No effect | n.d. | n.d. | 9.9 ± 1 µM | 16.3 ± 1.8 µM |

[a]Mean ± SD determined from three independent replicates
[b]Mean ± SD of independent replicates; $n = 5$ (Toyo), $n = 4$ (7-CN-7-C-Ino; 5-I-Tu), $n = 2$ (5-Br-Tu), $n = 2$ (tubercidin), $n = 1$ (sangivamycin), technical duplicates or triplicates each
[c]Mean ± SD of $n = 3$ independent replicates
[d]Mean ± SEM determined from independent replicates (Alamar blue cell viability assay): $n = 11$ (dipyridamole), $n = 8$ (5-I-Tu; 5-Br-Tu; sangivamycin in *pkar* KO), $n = 7$ (Toyo; sangivamycin in WT), $n = 6$ (7-CN-7-C-Ino)

---

(Fig. 3b, Table 1). The increased affinity was mainly due to gain in the entropic component of binding ($-T\Delta S = -2.3$ kcal mol$^{-1}$ versus $-0.9$ kcal mol$^{-1}$ for Toyo) (Fig. 3c). This surprising increase in activation potency upon introduction of a single structural modification did not, however, translate into increased potency in the in vivo kinase reporter assay (EC$_{50}$ 838 ± 139 nM), probably due to slower uptake or faster intracellular metabolism of the 7-deazainosine analogue. In a cell line with homozygous deletion of *PKAC3* and simultaneous RNAi depletion of *PKAC1/2* (Supplementary Fig. 4d), phosphorylation of the PKA reporter substrate VASP was undetectable with or without treatment by 7-CN-7-C-Ino (Supplementary Fig. 4c). This control fully corroborated the PKA specificity of the in vivo reporter assay based on VASP Ser-157 phosphorylation in trypanosomes. The cytotoxicity of 7-CN-7-C-Ino was drastically reduced compared to Toyo (>170-fold increase of EC$_{50}$ for cell viability, Table 1).

7-CN-7-C-Ino, Toyo, or 5-I-Tu did not bind to purified human PKARIα, whereas cAMP binding to the same PKARIα preparations ($K_D$ 28 nM, Supplementary Fig. 4f) was intermediate between the affinities of CNB-A and CNB-B that have been measured before separately[52]. Consistently, Toyo did not activate the mammalian PKARIα$_2$–2PKACα holoenzyme, even at concentrations 270-fold above the EC$_{50}$ for the *T. brucei* PKA holoenzyme (Supplementary Fig. 4g). As expected from the kinase assays (see Fig. 2b, d and Supplementary Fig. 3b), recombinant *T. brucei* PKAR did not bind cAMP (Supplementary Fig. 4h). In summary, a potent, highly specific, and nontoxic activator of the cAMP-independent trypanosomal PKA has been developed.

**Crystal structure of parasite PKAR with 7-CN-7-C-Ino**. To explore the molecular basis of binding and activation selectivity of *Trypanosoma* PKA for 7-deaza nucleoside analogues, we co-crystallized PKAR with 7-CN-7-C-Ino. The highest resolution X-ray diffraction data were obtained using truncated PKAR (200–503) of the related parasite *Trypanosoma cruzi* (Supplementary Fig. 5, see Supplementary Table 2 for data collection and refinement statistics) that is highly homologous in primary sequence (79% identity) and structural alignment to the *T. brucei* PKAR fragment. The protein adopts a dumbbell-like structure (Fig. 4a) consisting of two 8-stranded beta barrels (CNB-A and CNB-B) linked by an alpha helical element (R337 to N361, three helices). A 7-CN-7-C-Ino molecule can be clearly recognized on top of each beta barrel by the electron density in the Fo–Fc omit map (Fig. 4b, c). In bovine PKAR (PDB 1RGS), the CNB-A and CNB-B domains are linked via allosteric communication[30]. When a cAMP molecule enters the CNB-B of PKAR, it is capped by a hydrophobic residue (Y371) via π-stacking interaction. This event triggers the rupture of a salt bridge (E261–R366) and consequently destabilizes the holoenzyme conformation. Tryptophan W260 thereby moves 30 Å closer to CNB-A where it caps the second cAMP molecule in that binding site by an analogous π-stacking interaction. The conformational change releases an active C-subunit. The amino acids taking part in these sequential ligand binding events linked to holoenzyme activation are readily identified in 7-CN-7-C-Ino-bound *T. cruzi* PKAR by alignment with cAMP-bound bovine PKARIα (Fig. 4a, Supplementary Movie 1). In TcPKAR(200–503) the capping residues in CNB-A and -B are Y371 and Y483, respectively, while a salt bridge might be formed by the pair R476–E372. Despite only 35% sequence identity, the C$_\alpha$ alignment of both structures shows very high conservation (RMSD, root mean square deviation = 3.25 Å) including the fragmented αB/C helix (Supplementary Fig. 6, Supplementary Movie 2) that is key of the conformation change mediating allosteric activation of bovine PKARIα[30].

Both CNB-A and -B pockets are occupied by 7-CN-7-C-Ino with the ribose moiety in a position very similar to the position of the ribose moiety of cAMP bound to bovine PKARIα (Fig. 4a,

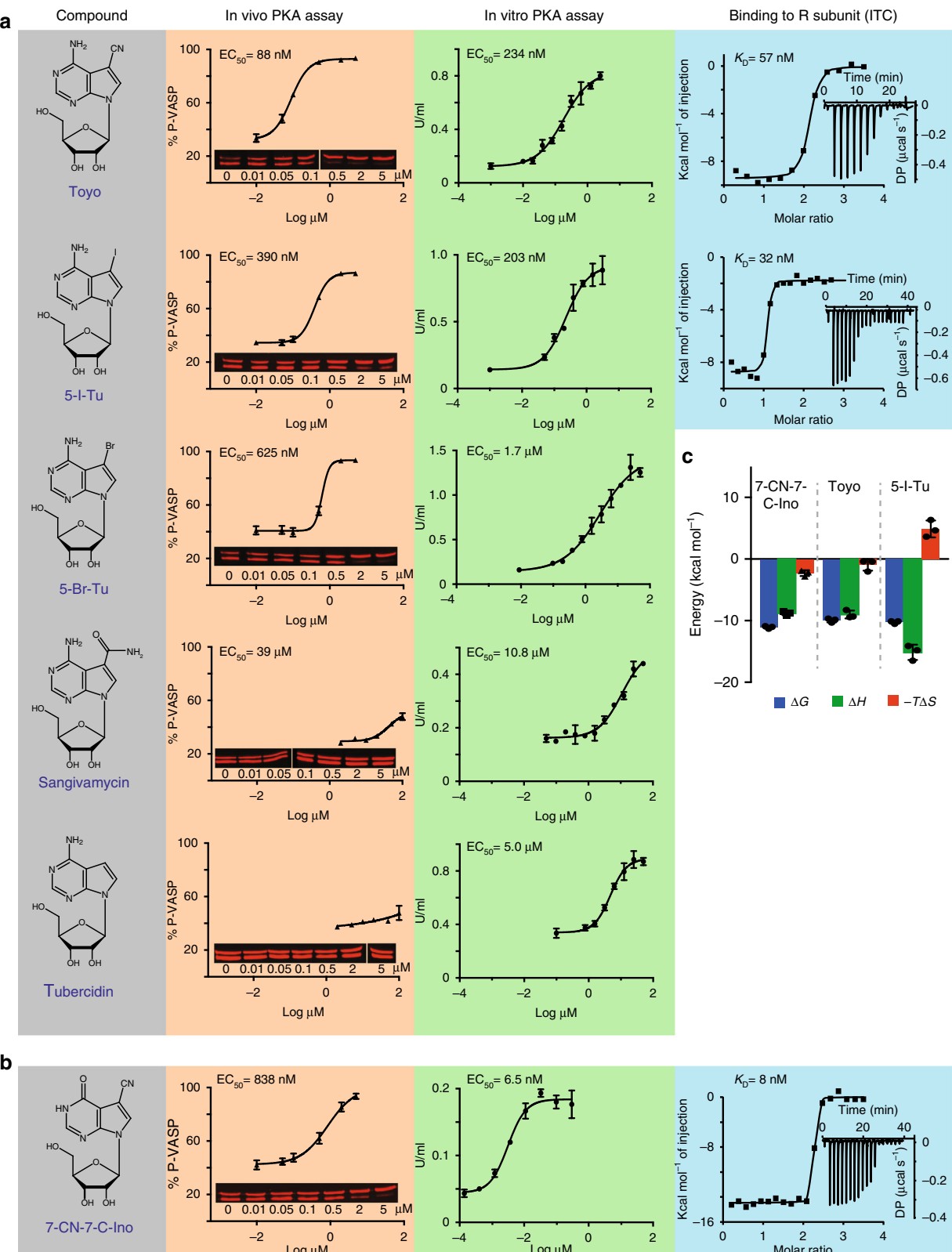

Supplementary Movie 1). The ribose moiety docks to the CNB-A site by donating two hydrogen bonds (hydroxyl groups O3 and O5) to E312 (Fig. 4d, Supplementary Movie 3). E310 contacts the O2 hydroxyl group and at the same time brings Y371 in close proximity to make a π-stacking interaction with the hypoxanthine moiety of the ligand. The free electron pair in the cyano moiety receives a hydrogen bond from the side chain of K294,

while Y300 additionally shields the ligand from the solvent. In the CNB-B site (Fig. 4e, Supplementary Movie 4), E436 interacts with the 3′ and 5′ hydroxyl groups of the ribose ring. Y483 acts as a hydrophobic cap that seems to be positioned via a water-mediated (w07) hydrogen bonding network formed by the main chain carbonyl group of H431, the side chain of E434 and the O2 hydroxyl group from the ribose. The keto group at position 6 of

**Fig. 3** Activators of trypanosome PKA. **a** Hit compounds with their $EC_{50}$ of in vivo PKA activity (VASP reporter assay, mean ± SD, $n = 3$ independent replicates, representative western blots as inset) and their $EC_{50}$ of in vitro kinase activity (kemptide phosphorylation by *T. brucei* PKAR-PKAC1 holoenzyme expressed in *L. tarentolae*). A representative dose response for Toyo ($n = 5$), 5-I-Tu ($n = 4$), 5-Br-Tu ($n = 2$), tubercidin ($n = 2$), and sangivamycin ($n = 1$) is shown with SD of technical duplicates or triplicates. Binding parameters to *T. brucei* PKAR(199–499) expressed in *E. coli* were determined by isothermal titration calorimetry (ITC). The power differential (DP) between the reference and sample cells upon injection was measured as a function of time (inset). The main plot presents the total heat exchange per mole of injectant (integrated peak areas from inset) as a function of the molar ratio of ligand to protein. One out of three independent replicates is shown. **b** Data for 7-CN-7-C-Ino as in **a**; for number of independent replicates see Table 1. The $EC_{50}$ and $K_D$ values given in **a** and **b** are rounded values from Table 1. **c** Thermodynamic signature ($\Delta G$ in blue, $\Delta H$ in green, and $-T\Delta S$ in red) compiled from ITC measurements (mean ± SD of $n = 3$ independent replicates) in **a** and **b**. Source data are provided as a Source Data file

the purine ring accepts a hydrogen bond from the main chain nitrogen atom of Y486, while the secondary amine in position 1 acts as a donor to the carbonyl of K482 (Fig. 4e, Supplementary Movie 4). In the CNB-B site, the cyano group does not engage in hydrophilic interactions but instead is sandwiched between the side chains of Y485 and Y486. A sequence alignment of kinetoplastid and mammalian PKAR (Fig. 4f, g) shows that alanine residues in both pockets (A202 and A326) of bovine PKAR are substituted by glutamates in the kinetoplastid PKAR orthologues. The positively charged arginine residues responsible for neutralizing the cyclic phosphate of cAMP (R209 and R333 in bovine PKAR) are substituted in kinetoplastid PKAs by neutral amino acids with shorter side chains (V, T, A, or N). Thus, few substitutions in a highly conserved signalling protein are hallmarks of a ligand selectivity switch from cAMP to 7-CN-7-C-Ino.

**Molecular docking of 7-deaza nucleosides**. For the series of 7-deaza nucleoside analogues, we found a good correlation between computational docking to the TcPKAR structure and the potency of activation of the purified kinase (Supplementary Fig. 7 and Table 1). Re-docking of the co-crystallized compound 7-CN-7-C-Ino to the CNB-A and -B sites of TcPKAR gave RMSD values of 0.266 and 0.252, respectively, for the best poses. Using the Glide E-model (GE) scoring system, the best docking poses of the compounds were ranked for the A site (7-CN-7-C-Ino > Toyo > 5-Br-Tu > 5-I-Tu > sangivamycin > Tu) and for the B site (7-CN-7-C-Ino > Tu > Toyo > 5-Br-Tu > 5-I-Tu > sangivamycin). Interestingly, Tu docks very well in the B site but is a weak activator (see Discussion). The cyano-, iodo-, or bromo- groups at position 5 of all other tubercidin analogues are accommodated in a hydrophobic pocket formed by the side chains of V444, V489, Y485, and Y486, the latter three being part of the αD helix (Supplementary Fig. 7c). 7-CN-7-C-Ino forms two additional hydrogen bonds (K482/O and Y486/N) with the αD helix compared to Toyo, correlated with a 7-fold higher affinity and 36-fold higher activation potency (Table 1).

**PKA signalling and targets in trypanosomes**. PKA downstream signalling components and targets are so far completely unexplored in trypanosomes. To probe PKA target phosphorylation events, we first used a phospho-specific PKA substrate antibody detecting the phosphorylated consensus PKA sites RXXS*/T*. A 3–4-fold increase in global RXXS*/T* site phosphorylation was observed by western blotting after 10 min of Toyo or 7-CN-7-C-Ino treatment in wild type but not in Δpkar/Δpkar cells (Fig. 5a). Phosphoproteome analysis under these conditions (15 min ± 7-CN-7-C-Ino) showed 642 significantly (FDR ≤ 0.05, $s_0 = 2$) upregulated phosphosites, mostly containing PKA motifs (77%) (K/R-X-X-S/T, K/R-X-S/T)[5], whereas these PKA motifs were underrepresented (19%) in the 84 downregulated phosphosites (Fig. 5b, c; Supplementary Data 1). The frequency distribution of PKA motif subsets in the 7-CN-7-C-Ino-induced *T. brucei*

phosphoproteome matches closely with that observed for human PKA motifs (Fig. 5c). An unbiased motif discovery algorithm confirmed enrichment of the same PKA motifs among the upregulated phosphosites (Supplementary Fig. 8a, b). Gene ontology (GO) enrichment analysis predicts functions of PKA in posttranscriptional regulation of gene expression, dynamics of cytoskeletal and organellar structures, signalling, and cell division and cytokinesis (Supplementary Fig. 8c).

We also explored changes in cellular protein abundance following PKA activation at later time points (6 and 12 h) by label-free quantitative proteomics. No significantly regulated proteins were detected 6 h post induction and only 14 proteins 12 h post induction in wild type cells (≥1.5-fold difference at $p \leq 0.05$, Supplementary Fig. 9), including PKAC1/2 and metacaspase 4 (MCA4)[53]. The Δpkar/Δpkar mutant trypanosomes served as negative control. As expected, PKAC1/2 levels decreased rapidly, resulting from ligand-induced dissociation of the holoenzyme complex and instability of the free C subunit, as reported for mammalian PKA[37]. MCA4, for which an antibody was available, informed on differential effects of 7-CN-7-C-Ino and Toyo: whereas both inducers decreased MCA4 abundance in wild type but not in Δpkar/Δpkar cells 12 h post induction, only Toyo elevated MCA4 at later time points in a PKA-independent fashion (Fig. 5d). This is compatible with observed PKA-independent effects of Toyo on growth (Table 1) due to multiple cellular targets. 7-CN-7-C-Ino, on the other hand, does not produce these PKA-independent effects on growth, viability, or visible phenotype. This compound is therefore proposed as novel and specific tool to study in vivo the essential processes regulated by PKA in trypanosomes.

## Discussion

Functionally and biochemically uncharacterized PKAs are regularly annotated as cyclic AMP-dependent protein kinases due to high conservation in the eukaryotic kingdom. Here, we have identified novel potent activators of cAMP-independent trypanosome PKA and unambiguously show that ligand selectivity has evolved away from cAMP. This challenges the current view that all PKA orthologues are cAMP-dependent and explains why earlier attempts to detect cAMP-stimulated kinase activity[23,24] or binding of cAMP to recombinantly expressed domain fragments of trypanosomal PKAR orthologs[22,27] remained negative. The CNB domain seems to be a more versatile ligand-binding domain, not limited to cyclic nucleotides, as reported for some distant CNB family members in prokaryotes[3,4]. We expect that systematic surveys will identify novel eukaryotic CNB domain binding specificities, not only in phylogenetically distant trypanosomatids.

Several conserved amino acids in the CNB domains of PKAR from *T. brucei* and related kinetoplastids depart from the consensus (see Fig. 4f, g), as noted earlier[22,27,31]. Yet, the amino acid sequence alone did not allow the prediction of altered ligand specificity. For example, the consensus arginine interacting with the exocyclic phosphate of cAMP in almost all PKAs (R333 in

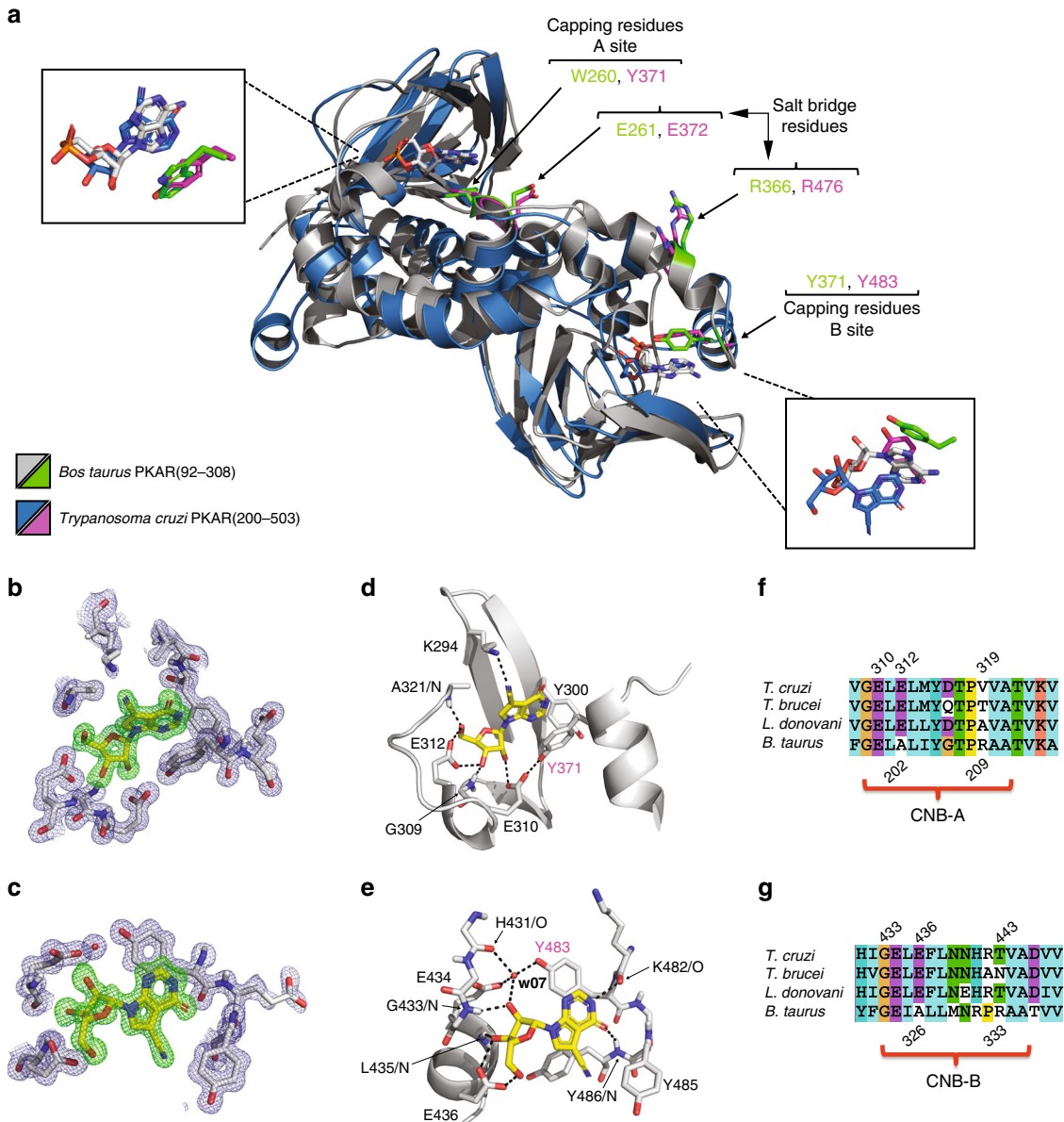

**Fig. 4** Co-crystal structure of trypanosome PKAR with 7-CN-7-C-Ino. **a** Structural alignment of *T. cruzi* PKAR(200–503) (chain representation in blue) and *Bos taurus* PKARIα(92–308) (PDB 1RGS, chain representation in grey). The two capping residues and the salt bridge pair are highlighted by their carbon atoms colour-coded as green and magenta in the *B. taurus* and *T. cruzi* PKAR structures, respectively. The overlay of ligand poses is shown in the blow-ups and the π-stacking interactions in both sites for both proteins are highlighted. **b**, **c** Fo-Fc (3σ, green) and 2Fo-Fc (1σ, blue) maps of TcPKAR CNB-A and CNB-B, respectively, showing 7-CN-7-C-Ino modelled to fit the electron densities. **d**, **e** Hydrogen bonding network (black dashed lines) of 7-CN-7-C-Ino bound to TcPKAR(200–503) CNB-A and CNB-B, respectively. The capping residues in CNB-A (Y371) and CNB-B (Y483) are labelled in magenta. 3D versions of **a**, **d**, and **e** are available as Supplementary Movie 1, 3, and 4, respectively. **f**, **g** Sequence alignment of PKAR in CNB-A and CNB-B, respectively, of representative kinetoplastid parasites with a mammalian PKAR (*B. taurus* PKARIα) as reference. Numbering refers to *T. cruzi* (top) and *B. taurus* (bottom), respectively

bovine RIα) is one of the key replacements in kinetoplastid PKAR, but the mammalian RIα mutant of that residue retained cAMP activation at an only 5-fold decreased EC$_{50}$[54]. Our co-crystal structure shows that glutamate residues (E312/E436) present in both CNB pockets of TcPKAR may clash with the negatively charged phosphate of cAMP (Supplementary Movie 5) but strengthen the interaction with the ribose moiety of 7-CN-7-C-Ino. In addition, an important role of the αD helix, an extra helix only present at the C-terminus of trypanosome PKAR is suggested. These features are conserved in the *Trypanosoma* species *T. brucei* and *T. cruzi* that share the unusual ligand binding. The high affinity of the 7-deaza nucleoside analogues can

be explained by interactions with side groups on the purine ring. For example, a hydrogen bond donor–acceptor pair (Y486/N–K482/O) from the backbone of the trypanosome-specific αD helix (see Fig. 4e) favours the interaction with a ligand having an acceptor–donor pair at positions 1 and 6 of the purine ring. This is the case for the hypoxanthine-like purine ring in 7-CN-7-C-Ino (Fig. 4e, Supplementary Fig. 7b) but not the adenine-like purine ring in toyocamycin. In silico docking shows that the bulky cyano-, iodo-, or bromo- side groups fill a small hydrophobic pocket formed at the interface between the beta barrel (V444) and the αD helix (V489, Y485, Y486) in CNB-B, and correctly predicts the relative affinities of these analogues (Supplementary

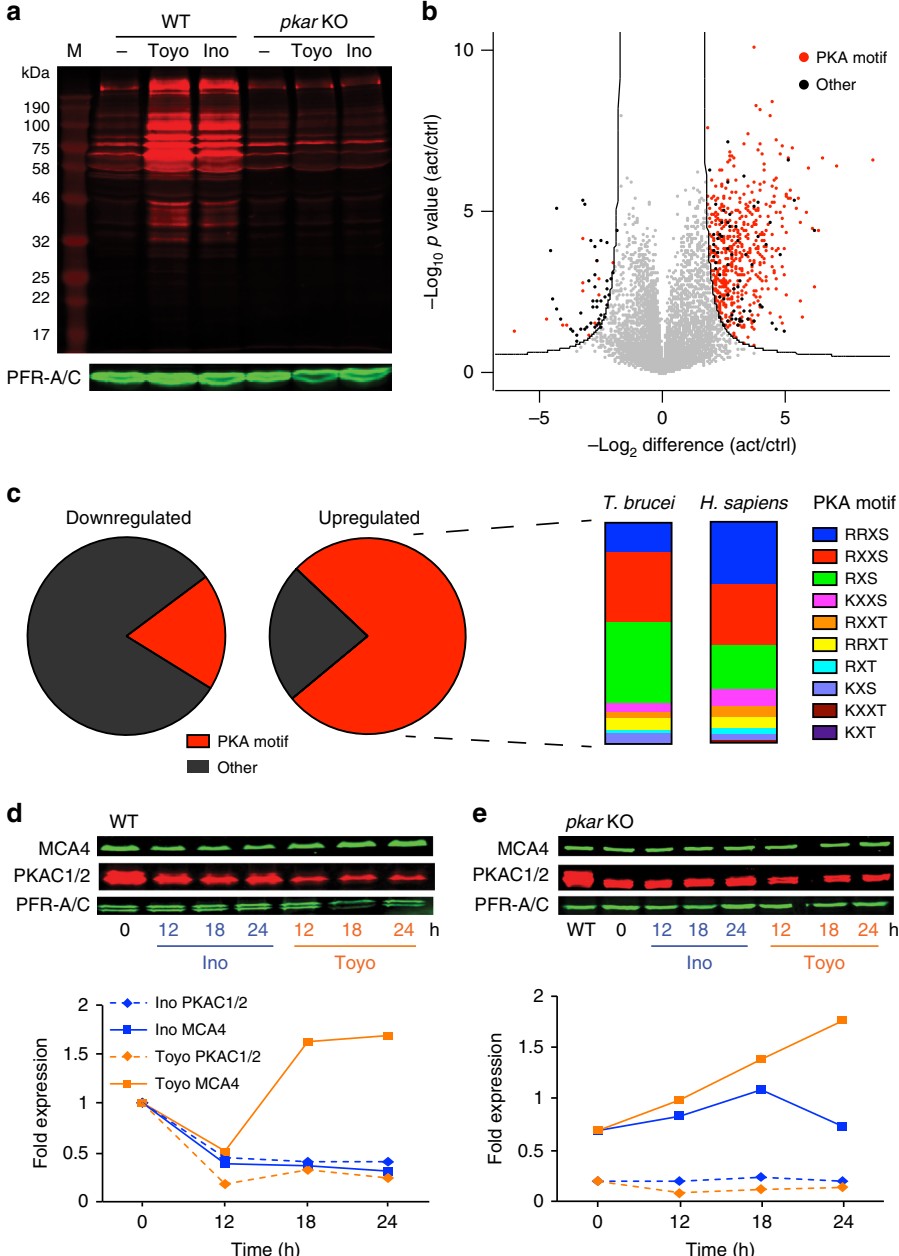

**Fig. 5** Target phosphorylation and expression changes. **a** Global display of proteins phosphorylated at RXXS/T sites. Wild type (WT) or Δ*pkar*/Δ*pkar* (*pkar* KO) cells were treated or not (−) with 2 μM Toyo or 7-CN-7-C-Ino (Ino) for 10 min and lysates were subjected to western blotting with anti-phospho-RXXS*/T* and anti-PFR-A/C as loading control. M: protein molecular weight marker. **b** Volcano plot representation of phosphopeptides quantified by label-free phosphoproteome analysis. Phosphopeptides are plotted according to p-value and fold change caused by treatment of *T. brucei* WT cells with 7-CN-7-C-Ino (8 μM, 15 min, $n = 4$ independent experiments) in comparison to untreated cells ($n = 4$ independent experiments). Phosphopeptides that change significantly in abundance (FDR ≤ 0.05, $s_0 = 2$) and contain phosphosites matching PKA consensus motifs (R/K-X-X-S/T, R/K-X-S/T)[5] are shown as red dots; significantly changed phosphopeptides without PKA consensus motifs are shown in black. **c** Pie charts showing the fraction of PKA consensus motifs (red) within the downregulated ($n = 84$, left) or upregulated ($n = 642$, right) phosphosites. The frequency of individual subsets of PKA motifs (upregulated) was compared to the human PKA site motif frequency retrieved from the PhosphoSitePlus database (https://www.phosphosite.org). **d** Abundance changes of metacaspase 4 (MCA4) and PKAC1/2 in WT or **e** *pkar* KO cells after treatment with 4 μM 7-CN-7-C-Ino (Ino) or Toyo in a 24-h time course. Western blot signals were normalized to the loading control PFR-A/C; untreated WT cells were set to 1. Source data to **a** and **c**–**e** are provided as a Source Data file

Fig. 7c). The low potency of sangivamycin can also be explained by the incompatibility of its bulky and hydrophilic side group with the small hydrophobic pocket. Interestingly, the second highest affinity to the B-site is predicted for tubercidin, while the kinase activation potency of this ligand is second lowest. Tubercidin has no side group at position 7; its ribose ring can sample more conformations and therefore possibly score higher in

docking. The missing contact to the αD helix is however correlated with low potency. This suggests a role of this trypanosome-specific helix not only for binding but also for the conformational change that underlays the activation mechanism. Only recently, the critical residues for cyclic nucleotide selectivity (cAMP/cGMP) between PKA and PKG have been defined[52,55]. Mutations G316R/A336T in CNB-B convert hRIα from low to high cGMP

affinity. Of note, the residue corresponding to A336 in hRIα is V444 in TcPKAR and is part of the hydrophobic pocket formed between beta barrel and αD helix. This pocket accommodates the substituent at the C7 position of our activators and may determine their selectivity.

The structural alignment between TcPKAR and bovine PKARIα shows conservation of the general polypeptide chain folding and of the putative salt bridge and capping residues linked to the extensively studied allosteric activation mechanism in mammalian PKARIα[1,30]. The rapid decrease of PKAC1 levels upon treatment of cells with 7-CN-7-C-Ino (Fig. 5d, Supplementary Fig. 9a) indicates that kinase activation involves holoenzyme dissociation in vivo. The paradigmatic features of the signalling protein PKA seem to be conserved and adapted to different activating ligands by minor substitutions in the PBC in concert with the trypanosome-specific αD helix extension in CNB-B. It follows that some of the "orphan" CNB domains identified in eukaryotic genomes[2,31,56,57] may be regulated by unexpected and novel ligands. Such discoveries will be as insightful as was the discovery of ligands for other classes of orphan receptors, e.g. from the G-protein coupled receptor[58] or nuclear hormone receptor[59] families.

Our report on this unique cAMP-independent PKA may appear to question the role of cAMP as second messenger in trypanosomes. Homologs of other known effector proteins like Epac or cyclic nucleotide gated ion channels were not found in the *T. brucei* genome. Yet, cAMP and the enzymes for its production and degradation are present and play vital roles in cell division and host immunity subversion[12,17–20]. We have previously identified a number of potential cAMP response proteins (CARPs) by genome-wide RNAi library screening in *T. brucei*[29]. CARP1 has three CNB domains and the *T. cruzi* ortholog[60] binds cAMP. Interestingly, *CARP1* is exclusively found in kinetoplastid genomes and possibly coevolved with the ligand swap of PKA away from cAMP. There are also additional candidate genes with predicted CNB domains, some of which may be cAMP effectors[41,60]. Divergent CNB domains as found in cAMP-binding Popeye-domain proteins[61] might be present but not recognized due to limited homology. Together, the cAMP-independence of PKA in trypanosomes is well compatible with cAMP signalling in these organisms.

7-CN-7-C-Ino is a potent activator and has low toxicity in vivo, in contrast to the 7-deazaadenosine analogues. The latter compounds are known to have pleiotropic effects in various cell types by interfering with cellular processes involving nucleosides or nucleotides. This comprises inhibition of kinases and inhibition of DNA, RNA, and protein synthesis (reviewed in ref. [47]) and explains the broad antibiotic and antineoplastic activity profile. A possible reason for the reduced off-target effects of 7-CN-7-C-Ino might be lower in vivo stability of the drug. This should not affect the short term in vivo kinase reporter assay but reduce off-target effects on growth and viability. In trypanosomes, toxicity of the 7-deazaadenosine analogues is clearly not mediated by PKA, as we find no difference between WT and Δpkar/Δpkar cells. A base level of catalytic PKA function is essential for growth and viability of *T. brucei* (Supplementary Fig. 2). Therefore, efficient inhibition of PKA activation by trapping the catalytic subunits in inactive holoenzyme complexes would reduce the free kinase to lethal level. The challenge in view of possible antiparasitic drug development will be to turn our novel activators to PKA inhibitors similar to what had been achieved for mammalian PKA[62]. As this work shows, TbPKA is essential, druggable, and has a ligand specificity differing from PKAR of the mammalian host. Due to conservation of the substitutions in the CNBs in different kinetoplastid PKAs, such inhibitors may also target the orthologues of related kinetoplastid pathogens like *Leishmania*.

With 7-CN-7-C-Ino, we contribute a potent chemical tool that will spur further investigations on the cellular PKA targets and physiological functions in trypanosomes. Our 7-CN-7-C-Ino-induced phosphoproteome returned a surprising number of highly upregulated P-sites, mostly (77%) within PKA consensus motifs. The GO terms most enriched among the target proteins are related to posttranscriptional control of gene expression and signalling. PKA phosphorylates directly or indirectly 17 other kinases, indicating interconnection with other signalling pathways. Participation in posttranscriptional control is not surprising as many RNA binding proteins and regulators of RNA stability or translation are regulated by phosphorylation in many organisms[63]. In trypanosomes, gene expression is almost exclusively controlled at the posttranscriptional level[64]. Therefore, PKA that participates in transcriptional regulation in other eukaryotes, may be redirected to the targets that exert gene expression control in trypanosomes. The cytokinesis phenotypes observed upon genetic perturbation of TbPKA (Supplementary Fig. 2 and ref. [35]) correspond well to enriched GO terms related to cell division and cytokinesis. The enrichment for cytoskeletal structures and the flagellum perfectly correspond to the subcellular localization of PKAR in the flagellum[65], a motility phenotype[65] and cytokinesis, a process that depends on the flagellum[66]. Surprisingly, the quantitative proteome analysis upon induction with 7-CN-7-C-Ino returned only 14 proteins significantly regulated in abundance after 12 h. Among them, downregulation of MCA4, an unconventional metacaspase regarded as pseudopeptidase[53] is interesting, since *T. brucei* mutants with homozygous deletion of *MCA4* have reduced virulence in animal infections[53].

The in vivo pathways and mechanisms upstream of PKA in trypanosomes are completely elusive to date. Is trypanosome PKA the effector of an unknown alternative second messenger? The direct activators of *T. brucei* PKA identified by this work include tubercidin, toyocamycin, and sangivamycin, natural antibiotics that are secondary metabolites of *Streptomyces* strains[67]. However, it seems unlikely that 7-deazapurines are the endogenous PKA ligands, since no homologs of the *Streptomyces* genes encoding enzymes required for synthesis of the precursor preQ0[68] were detected in the *T. brucei* genome. Of course, an alternative route of synthesis of a 7-deazapurine cannot be excluded. As we find indirect activation of *T. brucei* PKA by dipyridamole in live cells, this antimetabolite may perturb the parasite's nucleoside metabolism, e.g. by blocking uptake of adenosine[69]. Nucleoside-related metabolites or by-products of purine salvage may have adopted the role of second messenger-like molecules targeting PKA in trypanosomes. The identity of the physiological PKA ligand and the respective pathway is an exciting line of current research.

## Methods

**Trypanosome culture conditions**. Bloodstream forms of the monomorphic *Trypanosoma brucei brucei* strain Lister 427, variant MiTat 1.2[70], were cultivated at 37 °C and 5% $CO_2$ in modified HMI-9 medium[14] supplemented with 10% (v/v) heat-inactivated fetal bovine serum (FBS). Cell lines 13–90[71] or 1313–514[72] expressing T7 polymerase and Tet repressor were kept under continuous selection with 2.5 μg ml⁻¹ G418 and 5 μg ml⁻¹ hygromycin B or 0.2 μg ml⁻¹ phleomycin and 2 μg ml⁻¹ G418, respectively.

**Immunopurifications**. PTP purification was performed according to the protocol of Schimanski et al.[73] with a few modifications. Briefly, trypanosomes expressing PTP-tagged PKAR from the endogenous locus were lysed in PA-150 buffer (w/o DTT, supplemented with Complete Mini EDTA-free protease inhibitor cocktail (Roche) and 25 μg ml⁻¹ pepstatin A) by three sonication cycles 30 s each with the Bioruptor device (Diagenode) at low power. After centrifugation (20 min, 20,000 × g, 4 °C), the cleared supernatant was incubated with IgG beads (pre-equilibrated with PA-150 buffer) for 4 h to overnight by overhead rotation at low speed. Two washes with PA-150 buffer and one with PBS were followed by elution of proteins bound to the IgG beads by incubation with 2× Laemmli sample buffer (125 mM Tris, pH 6.8, 4% (w/v) SDS, 20% (v/v) glycerol, 10% 2-mercaptoethanol, 0.02% (w/v) bromophenol blue) for

5 min at 95 °C. Immunoprecipitation of Ty1- or HA-tagged PKA subunits was performed by incubation of trypanosomes in lysis buffer (50 mM Tris, pH 7.2, 2 mM EGTA, 150 mM NaCl, 0.2% NP-40, 1 mM NaVO$_4$, 0.5% aprotinin, 2 µg ml$^{-1}$ leupeptin, 1 mM PMSF) for 10 min on ice with subsequent clearing of the supernatant by centrifugation (20 min, 20,000 × g, 4 °C). The cleared lysate was incubated with the respective epitope-tag antibody coupled covalently (using the cross-linker DMP (Thermo Scientific) according to the manufacturer's instructions) or non-covalently to protein G sepharose beads (Amersham Pharmacia) for 1 h to overnight.

**In vivo PKA reporter assay**. Trypanosomes were harvested (10 min, 1400 × g, 37 °C) and resuspended in HMI-9 medium (pre-heated to 37 °C) to a density of 5 × 10$^7$ cells ml$^{-1}$. After a 5–10 min recovery at 37 °C with mild shaking, test compounds or solvent were added to the cell suspension followed by careful mixing. After incubation at 37 °C for the specified time period, cells were lysed with 6× Laemmli sample buffer preheated to 95 °C and incubated for 5 min at 95 °C. Incubation of trypanosomes at this density for up to 30 min had no effect on VASP phosphorylation (Supplementary Fig. 3g). GraphPad Prism 7.0 was used for EC$_{50}$ calculation by non-linear regression analysis using an equation for a sigmoidal dose–response curve with variable slope. The same software was used for visualization of all graphs and bar charts.

**Inhibitors and nucleoside analogues**. PKI(5–24) and myr-PKI(14–22) were obtained from Biomol. CpdA[19] (now renamed as NPD-001)[74] was synthesized by Geert-Jan Sterk, Mercachem. Dipyridamole was obtained from Sigma-Aldrich, toyocamycin and sangivamycin from Berry & Associates. All other nucleoside and cyclic nucleotide analogues were obtained from Biolog GmbH, Bremen. 7-CN-7-C-Ino was synthesized according to Hinshaw et al.[75] with 125 mg (429.1 µmol) Toyo as starting material. By variation to the original protocol, the raw product was purified by reversed phase medium pressure liquid chromatography (MPLC). Briefly, the raw product was dissolved in 20 ml water, filtrated and applied to a Merck LiChroprep® RP-18 column (15–25 µM; 125 × 35 mm), previously equilibrated with water. The column was washed with water to remove excess of inorganic salts and hydrophilic impurities. Afterwards, 1% and 2% 2-propanol in water was used to elute the target compound. Product-containing fractions were concentrated by rotary evaporation under reduced pressure and subsequently freeze-dried to yield 89.34 mg (305.7 µmol) of 7-CN-7-C-Ino with a purity of 99.93% by analytical HPLC (ODS-A 120-11, RP-18 (YMC, Dinslaken, Germany); 250 × 4 mm; 9% acetonitrile, 20 mM triethylammonium buffer, pH 6.8; 1.0 ml min$^{-1}$; UV-detection at 265 nm).

***Leishmania tarentolae* expression system**. The *L. tarentolae* strain LEXSY T7-TR (Jena Biosciences) was cultivated at 26.5 °C in BHI medium supplemented with 10 µg ml$^{-1}$ hemin, 100 U L$^{-1}$ streptomycin and 100 mg L$^{-1}$ penicillin according to the protocols provided by Jena Biosciences. For maintenance of T7 polymerase and Tet repressor, 10 µg ml$^{-1}$ nourseothricin (NTC) and 10 µg ml$^{-1}$ hygromycin B were added to the medium. For co-expression of *T. brucei* PKAR-10 × His and Strep-PKAC1, the full length ORFs were amplified from genomic DNA using primers introducing the respective epitope tag and cloned into pLEXSY_I-ble3 and pLEXSY_I-neo3 (Jena Biosciences), respectively. Details on primer sequences and cloning strategies are available upon request. Cells transfected with both constructs were cultivated in the presence of 100 µg ml$^{-1}$ phleomycin and 50 µg ml$^{-1}$ G418. *L. tarentolae* cells at mid log phase (2–3 × 10$^7$ cells ml$^{-1}$) were induced with 10 µg ml$^{-1}$ tetracycline for 24 h. Lysis in 50 mM Tris, pH 7.4, 150 mM NaCl, 0.2% Triton X-100, 1 mM 2-mercaptoethanol was completed by a Dounce homogenizer. Tandem affinity purification of the holoenzyme complex was performed to guarantee subunit stoichiometry and highest purity: His-tag purification using Ni-NTA beads (Thermo Fisher Scientific) was followed by Strep-tag purification using gravity flow chromatography and StrepTactin sepharose beads (IBA), according to the manufacturers' instructions. The eluted fractions were pooled and dialyzed against the kinase storage buffer (20 mM MOPS, pH 7.0, 150 mM NaCl, 1 mM 2-mercaptoethanol).

**In vitro kinase assay**. A radioactive PKA kinase assay was performed according to Hastie et al.[76], using 100 µM Kemptide (LRRASLG) as kinase substrate and 100 µM ATP spiked with [γ-$^{32}$P] ATP to give 200–400 cpm pmol$^{-1}$. GraphPad Prism 7.0 was used for EC$_{50}$ calculation by non-linear regression analysis using an equation for a sigmoidal dose–response curve with variable slope. The same software was used for visualization of all graphs and bar charts.

**Binding studies by isothermal titration calorimetry**. N-terminally truncated *T. brucei* PKAR (aa 199–499) was cloned into pETDuet-1 (Novagene) with an N-terminal His$_6$-tag and expressed in *E. coli* Rosetta (DE3). The bacteria were grown in Luria–Bertani (LB) medium to OD$_{600}$ of ~0.4 at 37 °C, followed by overnight induction with 0.4 mM IPTG at 20 °C. Cells were harvested by centrifugation and lysed in a French Press. Protein purification was done by Ni-NTA affinity chromatography, followed by elution in 50 mM HEPES, pH 7.5, 50 mM NaCl, and 250 mM imidazole. Eluted protein (~7 mg ml$^{-1}$) was dialyzed overnight in 50 mM HEPES, pH 7.5, 50 mM NaCl, 1 mM DTT, and stored at −80 °C. Ligand-free protein for binding studies was prepared by adding solid urea to the protein

solution to a final concentration of 8 M. After 1 h at 25 °C, the solution was passed over a prepacked PD10 column (GE Healthcare) equilibrated with 8 M urea, 50 mM HEPES, pH 8.5, 50 mM NaCl. TbPKAR(199–499) was refolded overnight by dialysis against 50 mM Tris, pH 8.5, 240 mM NaCl, 10 mM KCl, 2 mM MgCl$_2$, 2 mM CaCl$_2$, 0.4 M sucrose, 1 mM DTT at 4 °C, followed by separation of monomers from aggregates by size-exclusion chromatography on a Superdex 200 Increase 10/300 GL column (GE Healthcare) equilibrated with 50 mM HEPES, pH 7.5, 50 mM NaCl, and 1% DMSO (buffer A). Eluted protein was diluted to 10 µM for ITC. The human PKARIα (full size) was expressed from the pETDuet-1 based plasmid 6H.tev/HsPKARα in *E. coli* Rosetta D3, prepared cAMP-free according to Buechler et al.[77] with the following modification: the refolding buffer was the buffer used for TbPKAR refolding. The purified protein was diluted to 5 µM. For binding assays, 13–19 injections of 2–3 µl were performed with a MicroCal PEAQ-ITC (Malvern) instrument. In each series, 100 µM of ligand (prepared using buffer A) was injected at 298 K into 5–10 µM protein freshly eluted from size exclusion chromatography. Binding constants and thermodynamic data were derived from best least square fit analysis, applying a model with two binding sites (performed with MicroCal PEAQ-ITC software).

**Quantitative phosphoproteomics**. For each sample, 6 × 10$^8$ *T. brucei* cells treated or not with 8 µM 7-cyano-7-deazainosine for 15 min were lysed in 300 µl 4% (w/v) sodium deoxycholate, 0.1 M Tris, pH 8.5 for 5 min at 95 °C (according to the protocol of Humphrey et al.[78]). Samples were sonicated using a Bioruptor (Diagenode) (high power, two cycles of 10 min each, 30 s on/off). Protein concentration was determined by BCA protein assay and samples were adjusted to equal concentrations. Sample preparation and mass spectrometry were exactly carried out as described by Humphrey et al.[78]. MaxQuant 1.5.2.8[79] was used to identify proteins and quantify by LFQ with the following parameters: Database, TriTrypDB-39_TbruceiTREU_AnnotatedProteins; MS tol, 10 ppm; MS/MS tol, 0.5 Da; Peptide FDR, 0.1; Protein FDR, 0.01 Min. peptide Length, 5; Variable modifications, Oxidation (M), Phosphorylation (STY); Fixed modifications, Carbamidomethyl (C); Peptides for protein quantitation, razor and unique; Min. peptides, 1; Min. ratio count, 2. For proteomic analysis, identified proteins were considered as statistically significant with FDR ≤ 0.05 and $s_0 = 1$ (two-sided Student's *T*-test adjusted for multiple comparisons by Benjamini–Hochberg correction, Perseus[80]). Phosphopeptide analysis was carried out in Perseus as suggested by Humphrey et al.[78]. The mass spectrometry phosphoproteomics data have been deposited to the ProteomeExchange Consortium via the PRIDE partner repository (https://www.ebi.ac.uk/pride/archive) with the dataset identifier PXD012245. GO enrichment analysis was performed in TriTrypDB with default settings and visualized in Revigo (http://revigo.irb.hr). The motif discovery tool MoMo (using the motif-x algorithm) implemented in the MEME suite (http://meme-suite.org/) was used for unbiased motif discovery in the phosphoproteome dataset with the *T. brucei* TriTrypDB-40_TbruceiTREU927 protein database as background. Enriched sequence logos were visualized using CLC Main Workbench 7 (https://www.qiagenbioinformatics.com/).

**Reporting summary**. Further information on experimental design is available in the Nature Research Reporting Summary linked to this article.

## Data availability
The coordinates of the *T. cruzi* PKAR crystal structure bound to 7-CN-7-C-Ino have been deposited in the Protein Data Bank under the code PDB 6FTF. The phosphoproteome and proteome datasets are available in the PRIDE partner repository with the dataset identifiers PXD012245 and PXD009073, respectively. Genome sequence and annotation information was obtained from TritrypDB (http://www.tritrypdb.org). Human PKA substrates and phosphorylation motifs were retrieved from PhosphoSitePlus database (https://www.phosphosite.org). Gene ontology (GO) enrichment analysis was visualized using Revigo (http://revigo.irb.hr). The motif discovery tool MoMo implemented in the MEME suite (http://meme-suite.org) was used for unbiased motif discovery in the phosphoproteome dataset. The source data underlying Figs. 1c, d, 2a–h, 3a–c, 5a, c–e, Table 1, and Supplementary Figs. 1b, 2, 3a–d, f–i, 4a–h, 8a–c, 9a–c are provided as Source Data file.

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

## Acknowledgements

The authors thank P. Bastin (Pasteur Institute, Paris), A. Bindereif (University of Gießen), C. Dehio (University of Basel), K. Gull (University of Oxford), F. Herberg (University of Kassel), J.H. Leonhardt (LMU Munich), J. Mottram (University of York), E. Pays (Free University of Bruxelles), U. Walter (University of Würzburg), for plasmids or antibodies. A. Mourão (Helmholtz Zentrum Munich) helped with crystal handling and diffraction and T. Wein (LMU Munich) advised on molecular docking. The authors thank U. Manzau and U. Havemann for expert technical assistance with nucleoside and nucleotide syntheses. This work was supported by the University of Munich, Grants DFG 1100/7-1 and BELSPO PAI 6/15 to M.B. and a fellowship from the Science without borders/CNPq program to Y.V.S.

## Author contributions

S.B., Y.V.S., F.S. and M.B. designed research. S.B., Y.V.S., S.K., G.B.G., T.K., J.P., C.B., F.S., J.-W.D. J.B. and I.F. performed research. R.S. wrote a script for analysis. H.-G.G. contributed reagents. S.B., Y.V.S., J.B., E.L., I.F., A.I. and M.B. analyzed data. S.B., Y.V.S. and M.B. wrote the paper.

## Additional information

**Competing interests:** F.S. and H.-G.G. are employed by the BIOLOG Life Science Institute that sells nucleosides and nucleotides that were used in this study or were developed for this study. The other authors declare no competing interests.

