## [Peer Review File · Nature Communications]

Reviewers' Comments:

Reviewer #1:

Remarks to the Author:

In this paper the authors provide convincing evidence that protein kinase A of the parasitic protozoan, *Trypanosoma brucei*, is refractory to activation by cAMP. They identify a potent non-physiological activator of PKA and provide a high resolution co-crystal structure (from the related *Trypanosoma cruzi*) that identifies the key residues that account for ligand binding. They propose that in trypanosomes PKA has evolved to respond to another (as yet unidentified) nucleoside-related endogenous signalling molecule.

The identification of cAMP-dependent protein kinase activity in trypanosomes and related trypanosomatid organisms (such as *Leishmania*) has proved elusive, despite the presence in the genome of protein kinase genes that encode PKA-like enzymes. The very high quality and extensive biochemical data presented in this manuscript provides a structural explanation for why *T. brucei* PKA (and other trypanosomatids) is unresponsive to cAMP (as propounded by Bubis et al., 2018 in *T. equiperdum*, ref 56) . The identification of 5-substituted tubercidin analogues as activators of *T. brucei* PKA hint at a novel mechanism for PKA activation in trypanosomes and provide some preliminary support for the concept that activation of PKA might be a drug target. Key questions that remain unanswered include; what is the physiological activator of trypanosome PKA? what are the substrates of PKA? what signalling pathway(s) is PKA involved in? A slightly more existential question is whether the trypanosome protein kinases under investigation are true PKA orthologues (should they be called PKA?). Until these questions are answered, it is difficult to predict the functional relevance of the apparent evolutionary divergence of the trypanosome PKAs.

Some specific comments

1. Line 93. There is an argument to be made that development of the parasite correlates with AMP (signalling), rather than cAMP (eg Ref 15, also Money et al., Nature)
2. Fig 1c labelling is confusing as Anti-PKAR is an epitope tag not an antibody to PKAR
3. Fig 1c – what is the evidence that the doublet bands are due to modification of PKAC1?
4. Line 178. Provide a reference for the absence of a check point for re-initiation of S phase
5. Line 211. VASP S157 is also phosphorylated by PDK1 and PKG (PMC3916352). What evidence is there that S157 is specific to PKA in trypanosomes? VASP can also be phosphorylated by AMPK, which is also influenced by hydrolyzable cAMP analogues.
6. Figure 2b. The % of P-VASP is quite high even at time 0 – this may influence the shift in % at different time points later. Is the total of VASP and P-VASP consistent through the time course? It looks somewhat variable eg 2b vs 2c.
7. Figure 2d. A useful control would be pre-treatment with myr-PKI to show that all VASP is non-phosphorylated.
8. Line 382. The crystal structure was obtained with the *T. cruzi* enzyme, yet all the rest of the biology/biochemistry was carried out with *T. brucei*. What evidence is there that PKA in *T. cruzi* PKA has the same biology?
9. The rationale for using 6 and 12 h time points for identifying the early changes in cellular protein abundance elicited by PKA activation is unclear. The experiment in figure 2 was 10 – 15 minute incubations with compound DIP. Why were these time points chosen.
10. Toyo treatment upregulated MC4 (called MCA4 in the original paper), yet what is the link to PKA?

Reviewer #2:

Remarks to the Author:

Bachmaier and colleagues present a comprehensive study of protein kinase A (PKA) activation in *Trypanosoma* species. In all eukaryotes, PKA activity is regulated by cyclic nucleotides, usually cAMP, that bind to an ancient cyclic nucleotide-binding (CNB) domain. While *Trypanosoma* PKA contains a CNB, the enzyme is not activated by cAMP. Starting from a known class of *Trypanosoma* PKA activators, the 7-deaza-adenosine (tubercidin) derivatives, the authors identify the inosine derivative 7-CN-7-C-Ino as a high-affinity binder and potent activator of *T. brucei* PKA both in vitro and in vivo. By analyzing the structure of 7-CN-7-C-Ino bound to a truncated regulatory subunit, PKAR(200-503), from *Trypanosoma*, they are able to identify the mutations and subtle conformational adjustments that lead to a change from cAMP to 7-CN-7-C-Ino activation in the parasite PKA. Through the structural analysis and by a wide array of functional assays, both in vitro and in vivo, the authors demonstrate convincingly that *Trypanosoma* PKA is refractory to activation by cAMP, but susceptible to activation by an array of nucleoside derivatives, 7-CN-7-C-Ino being the most potent.

The major claim of this manuscript, that *Trypanosoma* PKA is unique among its homologs by showing a clearly different activation mechanism, is supported by a convincing set of data. This novel insight will be of interest to a wide audience of cell biologists and parasitologists, in particular because the nucleoside activators described in the paper promise to become useful tools for studying cell signaling in *Trypanosoma*. The study offers much insight into PKA signaling in *Trypanosoma*, but falls short of identifying the physiological activators of *Trypanosoma* PKA.

Major comments:

1. none

Minor comments:

2. Line 138: There is a reference to Fig. 1b, but I did not see any reference to Fig. 1a; please check.
3. Line 201: How high are "very high" concentrations of cGMP? A number would help.
4. Fig. 3 and Table 1 show that some tested compounds are more potent PKA activators in vivo than in vitro, e.g. Toyo. Why is that? Are these nucleosides possibly metabolized to their active form as PKA activators? Can they be recovered from in vivo assays?
5. The crystal structure description in the text (lines 378 to 436, all in one paragraph) is almost excessively detailed and may be shortened and/or broken down to shorter sections for better readability.
6. A "helical segment R266-K250" is mentioned on line 403. Are the residue numbers correct? Please check.
7. Lines 989-990: Please change "S1 Table" to "S2 Table".
8. Table S2: Why is the data completeness only 82% over the whole data set and only 41% in the outer shell? I do not think this is a very serious problem, because the unbiased electron density shown in Supplementary Fig. 5c is convincing, but the lack of completeness should be justified, and it should be conceded that the effective resolution, as a consequence, is lower than the 1.092 Å stated in the manuscript. Was the unusually small test set with <2% of the diffraction data chosen because of the data incompleteness? Please explain. I suggest to replace the section header "Cell dimensions" with "Diffraction data" and to move the lines "R-meas (%)", "CC ½" and "Wilson B-factor (Å²)" up into this section, where they belong. The authors may want to replace the term "Redundancy" with the better term "Multiplicity". Finally, I note that the Ramachandran angles should add up to 100%.

Reviewer #3:

Remarks to the Author:

General comment.

This manuscript describes a very thorough and well-controlled set of experiments that clearly establish that cAMP is not the natural Trypanosome ligand for TbPKAR. It has however 2 shortcomings:

Novelty:

They clearly state:

"In bacterial transcription factors, some CNB domains can bind other ligands like heme in the case of the CO sensing transcription activator CooA (4) or chlorinated phenolic compounds in CprK, a member of the ubiquitous CRP77 FNR family of transcription activators (5)."

So, the concept that CNB domains can in fact bind non-cAMP ligands has been established, albeit in bacteria.

Discussion lines 540-541: « This challenges the current view that all PKA orthologs are cAMP-dependent » Surely that view was already challenged refs 4 & 5?

What then is the natural ligand?

The shortcoming of not knowing what the real Trypanosome ligand activates PKA is reflected both in the title:

"Nucleoside analog activators of cyclic AMP-independent protein kinase A of Trypanosoma »

And in the Discussion:

Lines 664-665: "Is trypanosome PKA the effector of an unknown alternative second messenger? »

Lines 677-678: "The identity of the physiological PKA ligand and the respective pathway will be an exciting future research question."

Line 330-334: "The evidence for a direct mode of action provided by in vitro kinase assays was further corroborated by measuring binding parameters between the activating compounds and purified N-terminally truncated *T. brucei* PKAR(199-499) expressed in *E. coli* (S4f Fig)."

It's a shame then that they didn't exploit binding to the CNB in TbPKAR to identify the natural in vivo ligand.

Minor points:

Lines 137-138: "PKAC1 was Ty1-epitope tagged in situ and the second PKAC1 allele was deleted to generate *T. brucei* cell line $\Delta c1/Ty1-C1$ (Fig 1b). » I think they mean PKAC2?

Lines 489-491: "Early changes in cellular protein abundances elicited by PKA activation were identified by label-free quantitative proteomics. » What is the logic of looking at protein abundance as a readout of PKA activity?

Lines 512-513: "Therefore, 7-CN-7-C-Ino has higher PKA specificity with respect to target protein regulation » Now we've switched from abundance to regulation.

Accession codes for phosphoproteome and proteome data can be found at the end of this document.

Reply to reviewers' comments:

Reviewer #1 (Remarks to the Author):

In this paper the authors provide convincing evidence that protein kinase A of the parasitic protozoan, *Trypanosoma brucei*, is refractory to activation by cAMP. They identify a potent non-physiological activator of PKA and provide a high resolution co-crystal structure (from the related *Trypanosoma cruzi*) that identifies the key residues that account for ligand binding. They propose that in trypanosomes PKA has evolved to respond to another (as yet unidentified) nucleoside-related endogenous signalling molecule.

The identification of cAMP-dependent protein kinase activity in trypanosomes and related trypanosomatid organisms (such as *Leishmania*) has proved elusive, despite the presence in the genome of protein kinase genes that encode PKA-like enzymes. The very high quality and extensive biochemical data presented in this manuscript provides a structural explanation for why *T. brucei* PKA (and other trypanosomatids) is unresponsive to cAMP (as propounded by Bubis et al., 2018 in *T. equiperdum*, ref 56). The identification of 5-substituted tubercicidin analogues as activators of *T. brucei* PKA hint at a novel mechanism for PKA activation in trypanosomes and provide some preliminary support for the concept that activation of PKA might be a drug target. Key questions that remain unanswered include; what is the physiological activator of trypanosome PKA? what are the substrates of PKA? what signalling pathway(s) is PKA involved in?

We fully agree with the importance of investigating (1) the identity of the physiological activator and (2) the downstream substrates and pathways. We reply specifically to (1) in the context of comments of reviewer 3. To address (2) and to increase the impact of our manuscript we have now performed a detailed phosphoproteome analysis in trypanosomes induced or not with the PKA activator 7-CN-7-C-Ino (new Fig. 5b, c and new Supplementary Fig. 7). The data are informative with respect to the predominant phosphorylation site motif, the classification of TbPKA as PKA ortholog and GO term enrichment of the targets. The phosphoproteome provides a valuable resource for further work on the downstream pathways. The results section has been adapted (lines 409-421). We also discuss the correspondence of target groups with the observed phenotypes upon RNAi-mediated PKA repression in this work and the literature (lines 568-574).

A slightly more existential question is whether the trypanosome protein kinases under investigation are true PKA orthologues (should they be called PKA?). Until these questions are answered, it is difficult to predict the functional relevance of the apparent evolutionary divergence of the trypanosome PKAs.

We made a case for the trypanosomal PKA to be a true orthologue, hence the gene name adopted. The arguments in the manuscript include overall sequence conservation of both catalytic and regulatory subunits, perfect structural alignment of the crystallized R subunit, PKA-specific kinase activity and conservation of residues known to be involved in the activation mechanism of other PKAs, except those residues defining the ligand specificity. We accept the comment of the reviewer that this is an "existential question". Therefore, we have now added two experiments: First we show that the *in vivo* reporter (VASP) assay shown to be PKA-specific in the mammalian system is also perfectly specific for TbPKA in trypanosomes (lines 304-310; new Supplementary Fig. 4c). Second, phosphoproteome analysis upon PKA induction returns exactly the same phosphorylation site motif distribution reported for mammalian PKA (lines 409-421; new Fig. 5b, c and new Supplementary Fig. 8). Additions to the discussion are found in lines 448-450 and 557-576.

Some specific comments

1. Line 93. There is an argument to be made that development of the parasite correlates with AMP (signalling), rather than cAMP (eg Ref 15, also Money et al., Nature We agree that we have not well covered this. The argument is more explicit now and more references have been added (lines 83-85).
2. Fig 1c labelling is confusing as Anti-PKAR is an epitope tag not an antibody to PKAR A misunderstanding was created as the lettering was not identical in Fig. 1c (Anti-PKAR) and the legend (anti-R). This has been corrected in the legend. The polyclonal PKAR-specific antibody was used to detect both the PTP-tagged R subunit and the endogenous PKAR.
3. Fig 1c – what is the evidence that the doublet bands are due to modification of PKAC1? MS data and phospho-site mutants in the PhD thesis of one of the authors (available online at <https://edoc.ub.uni-muenchen.de/3838/>) have confirmed phosphorylation as the cause of the PKAC1

doublet band, and genomic PCR has confirmed an allelic polymorphism for PKAR. As these data are completely irrelevant for the subject of the manuscript we would like to leave a minimal statement as “unpublished observation” just to explain the additional bands.

4. Line 178. Provide a reference for the absence of a check point for re-initiation of S phase
Unfortunately, we made a mistake in the text: the phenotype can indeed not be attributed to “absence of a check point for re-initiation of S phase” as we have no evidence for a block of mitosis. We have added appropriate references for “cytokinesis phenotypes” (lines 152-153).

5. Line 211. VASP S157 is also phosphorylated by PDK1 and PKG (PMC3916352). What evidence is there that S157 is specific to PKA in trypanosomes? VASP can also be phosphorylated by AMPK, which is also influenced by hydrolyzable cAMP analogues.

In Supplementary Fig. 3e we use the PKA-specific peptide inhibitor PKI as control that does not inhibit PDK1 and AMPK in mammalian cells. There is no PKG in trypanosome genomes. We cannot refute the argument though, that the PKI inhibitor profile might be species-specific. Therefore, we now add more definitive evidence by a genetic experiment: we have engineered a VASP reporter cell line with homozygous deletion of *PKAC3* and inducible RNAi of *PKAC1* and *PKAC2*. A triple knockout of all three PKA catalytic subunit genes would be impossible due to essentiality of *PKAC1/2* (see Supplementary Fig. 2). The result shown in the new Supplementary Figure 4c confirms that phosphorylation of our VASP reporter (at S157) is strictly PKA-specific in trypanosomes. The additional control is mentioned in results (lines 304-310) and described in more detail in the Supplementary Methods section and the legend to Supplementary Fig. 4c.

6. Figure 2b. The % of P-VASP is quite high even at time 0 – this may influence the shift in % at different time points later. Is the total of VASP and P-VASP consistent through the time course? It looks somewhat variable eg 2b vs 2c.

The basal VASP phosphorylation is indeed (biologically) variable from experiment to experiment; this may slightly change the value of an increase at later time points (reducing the dynamic range of the assay) but not affect the significance of any increases reported in the manuscript. The assay is ratiometric (as mentioned in lines 187-189), therefore the total of P-VASP and VASP does not affect the calculated % of P-VASP and a loading control is not required. This is one of the reasons why this reporter assay is technically very reliable.

7. Figure 2d. A useful control would be pre-treatment with myr-PKI to show that all VASP is non-phosphorylated.

Treatment with myr-PKI alone inhibits VASP phosphorylation to 3-10% depending on the inhibitor batch. Probably limiting permeability and low concentration of the myr-PKI in cell culture medium (cost!) explain this background. A more rigorous genetic experiment demonstrating PKA-specificity of the VASP phosphorylation assay *in vivo* is described in our reply to comment 5 above.

8. Line 382. The crystal structure was obtained with the *T. cruzi* enzyme, yet all the rest of the biology/biochemistry was carried out with *T. brucei*. What evidence is there that PKA in *T. cruzi* PKA has the same biology?

We have first crystallized the *T. brucei* PKAR protein, but the *T. cruzi* ortholog gave higher resolution upon co-crystallization with 7-CN-7-C-Ino. The *T. brucei* and *T. cruzi* PKAR are highly homologous in primary sequence, domain architecture, and structural alignment. All amino acids involved in ligand binding are identical. Structural alignment between TcPKAR and cAMP-bound bovine PKAR1 α resulted in major clashes of cAMP with the hallmark glutamates in the *T. cruzi* binding pockets (Supplementary movie 5). All evidences therefore suggest that the “biology of the enzyme” is the same, i.e. *T. cruzi* PKA is also cAMP-independent. We are aware of earlier reports claiming cAMP regulation of TcPKA, yet have not cited them as we have critically discussed these data already in Bachmaier, S., and Boshart, M. (2013, reference 27 in manuscript). We still think that technical problems account for those results completely incompatible with sequence and structure information. Binding assays by ITC (as in Supplementary Fig. 4h of the manuscript for the *T. brucei* protein) to show absence of cAMP binding were not possible as the apo-form of the *T. cruzi* protein is highly unstable (in spite of better crystallization properties with the 7CN-7-C-Ino ligand).

9. The rationale for using 6 and 12 h time points for identifying the early changes in cellular protein abundance elicited by PKA activation is unclear. The experiment in figure 2 was 10 – 15 minute incubations with compound DIP. Why were these time points chosen.

To monitor induced phosphorylation we have always used 10-15 min (e.g. for DIP, Toyo, or 7-CN-7-C-Ino in Fig. 2, Supplementary Fig. 4b, Fig. 5a, respectively) in agreement with published phosphoproteome analyses using various kinases and drug activators. The selected time points for measurement of changes in induced protein abundance (subsequent to phosphorylation, an independent experiment) were selected according to the time course of expression changes observed in other proteome studies in trypanosomes (see <http://tritrypdb.org>). Our results showed no protein

abundance changes at 6h (Supplementary Fig. 8b, c) and few changes at 12 h (Supplementary Fig. 8b, c) indicating that the choice of time points was appropriate to detect “early changes”.

10. Toyo treatment upregulated MC4 (called MCA4 in the original paper), yet what is the link to PKA? We unintentionally had changed the gene name, this has been corrected in the revised manuscript. MCA4 decreases at early times upon PKA activation (Toyo and 7-CN-7-C-Ino), the mechanism (direct or indirect) has not been investigated. We looked at this protein because it is also elevated at later time points by Toyo, but not by 7-CN-7-C-Ino. This second phase of regulation is PKA-independent and illustrates the higher specificity of PKA activation by 7-CN-7-C-Ino as compared to Toyo. Thereby we show that inducer optimization was successful and resulted in a more specific activator. We have now rephrased the results section on MCA4 (lines 430-437).

Reviewer #2 (Remarks to the Author):

Bachmaier and colleagues present a comprehensive study of protein kinase A (PKA) activation in Trypanosoma species. In all eukaryotes, PKA activity is regulated by cyclic nucleotides, usually cAMP, that bind to an ancient cyclic nucleotide-binding (CNB) domain. While Trypanosoma PKA contains a CNB, the enzyme is not activated by cAMP. Starting from a known class of Trypanosoma PKA activators, the 7-deaza-adenosine (tubercidin) derivatives, the authors identify the inosine derivative 7-CN-7-C-Ino as a high-affinity binder and potent activator of *T. brucei* PKA both in vitro and in vivo. By analyzing the structure of 7-CN-7-C-Ino bound to a truncated regulatory subunit, PKAR(200-503), from Trypanosoma, they are able to identify the mutations and subtle conformational adjustments that lead to a change from cAMP to 7-CN-7-C-Ino activation in the parasite PKA. Through the structural analysis and by a wide array of functional assays, both in vitro and in vivo, the authors demonstrate convincingly that Trypanosoma PKA is refractory to activation by cAMP, but susceptible to activation by an array of nucleoside derivatives, 7-CN-7-C-Ino being the most potent.

The major claim of this manuscript, that Trypanosoma PKA is unique among its homologs by showing a clearly different activation mechanism, is supported by a convincing set of data. This novel insight will be of interest to a wide audience of cell biologists and parasitologists, in particular because the nucleoside activators described in the paper promise to become useful tools for studying cell signaling in Trypanosoma. The study offers much insight into PKA signaling in Trypanosoma, but falls short of identifying the physiological activators of Trypanosoma PKA.

Major comments:

1. none

Minor comments:

2. Line 138: There is a reference to Fig. 1b, but I did not see any reference to Fig. 1a; please check. We refer to Fig. 1a in the introduction (see line 104).

3. Line 201: How high are “very high” concentrations of cGMP? A number would help. In Supplementary Fig. 3b we determined $\geq 100\mu\text{M}$. We also cite Shalaby et al. (reference 23 in manuscript), although they report $\geq 20\mu\text{M}$, a value we were not able to reproduce upon multiple repetitions. We have now added $\geq 100\mu\text{M}$ to the text (lines 175-176).

4. Fig. 3 and Table 1 show that some tested compounds are more potent PKA activators in vivo than in vitro, e.g. Toyo. Why is that? Are these nucleosides possibly metabolized to their active form as PKA activators? Can they be recovered from in vivo assays? We had attributed these relatively small differences to uptake, intracellular accumulation or metabolism of the drugs. For Toyo and 5-Br-Tu the higher *in vivo* potency in trypanosomes could theoretically be due to unknown metabolism to an active form, as suggested by the reviewer, but in light of the much higher absolute *in vitro* potency of the optimized 7-CN-7-C-Ino we did not investigate the other compounds any further. A comprehensive analysis of possible *in vivo* metabolites of the drugs would be also be beyond the scope of this research.

5. The crystal structure description in the text (lines 378 to 436, all in one paragraph) is almost excessively detailed and may be shortened and/or broken down to shorter sections for better readability. We agree with the reviewer that this section can be shortened; we have reduced from 1020 to 869 words (15%) and divided in two sections.

6. A “helical segment R266-K250” is mentioned on line 403. Are the residue numbers correct? Please check.

Thanks to the reviewer for indicating this typo: the helical segment is R226-K250, indeed.

7. Lines 989-990: Please change "S1 Table" to "S2 Table".
done

8. Table S2: Why is the data completeness only 82% over the whole data set and only 41% in the outer shell?

The low completeness of 82% is a result of low completeness for the high resolution shells and not a result of low completeness for the entire resolution range. The data are >98% complete in the range of 40-2.3 Å and 94.1% complete in the 1.64-1.46 Å resolution shell (see screen shot from data processing using XDS below). Given that the inclusion of low completeness data with significant signal-to-noise improves refinement (more data points), we have decided to keep these reflections in the data set rather than to exclude them.

SUBSET OF RESOLUTION LIMIT	INTENSITY OBSERVED	DATA WITH UNIQUE	SIGNAL/NOISE POSSIBLE	>= -3.0 AS FUNCTION OF DATA	R-FACTOR observed	R-FACTOR COMPARED expected	I/SIGMA	R-meas.	CC(1/2)	Anomal. Corr.	SigAno.	Nano	
3.27	35425	9745	9869	98.7%	3.1%	3.0%	35353	39.32	3.7%	99.8*	14*	0.946	4688
2.31	62883	17668	18818	98.1%	3.5%	3.4%	61960	38.83	4.1%	99.8*	7	0.890	8564
1.89	82263	22570	23186	97.3%	4.5%	4.5%	82119	23.99	5.3%	99.7*	3	0.827	11813
1.64	91830	26516	27499	96.4%	7.4%	7.7%	98744	14.07	8.8%	99.2*	-1	0.779	12875
1.46	106931	29314	31166	94.1%	13.7%	14.4%	106516	8.67	16.1%	98.0*	1	0.779	14223
1.34	105951	31014	34503	89.9%	24.2%	24.7%	105373	4.89	28.8%	93.0*	0	0.774	14932
1.24	110510	31147	37347	83.4%	41.2%	40.7%	109963	3.01	48.6%	84.1*	1	0.801	15083
1.16	101324	29652	40371	73.4%	57.7%	57.1%	100316	2.04	68.4%	71.6*	-3	0.745	14105
1.09	51362	17564	42807	41.9%	74.7%	75.9%	48348	1.39	89.5%	60.6*	-1	0.713	6741
total	746879	215190	264766	81.3%	5.6%	5.6%	740692	11.28	6.6%	99.9*	1	0.795	10224

NUMBER OF REFLECTIONS IN SELECTED SUBSET OF IMAGES	751790
NUMBER OF REJECTED MISFITS	4897
NUMBER OF SYSTEMATIC ABSENT REFLECTIONS	0
NUMBER OF ACCEPTED OBSERVATIONS	746893
NUMBER OF UNIQUE ACCEPTED REFLECTIONS	215194

I do not think this is a very serious problem, because the unbiased electron density shown in Supplementary Fig. 5c is convincing, but the lack of completeness should be justified, and it should be conceded that the effective resolution, as a consequence, is lower than the 1.092 Å stated in the manuscript.

We agree with the reviewer that the effective resolution is rather 1.15-1.2 Å given the lower completeness. We have added a sentence in the Methods section explaining this (lines 862-865) and have removed the resolution from the abstract. However, none of the conclusions of the manuscript depends on whether the structure is of 1.1, 1.15 or 1.2 Å resolution.

Was the unusually small test set with <2% of the diffraction data chosen because of the data incompleteness? Please explain.

For the Rfree value to be of significance the test set has to include >500 reflections, the actual percentage of reflections chosen is not of importance (see screen shot from CCP4 manual below). For the data set presented in our manuscript, the 2% test set represents >5000 reflections, which is 10X more than the recommended minimum. Including more reflections would be a waste as they do not increase the statistical significance of the calculated Rfree value and are excluded from the refinement and thus do not contribute to refining the parameters of the structure.

Choosing a FreeR fraction

It is important to choose a fraction that is large enough so that the statistics are sensible (at least 500 reflections seems to be the consensus at the moment), but small enough so that as many reflections as possible are still used for the refinement. This is of course always true, whichever philosophy is chosen for the selection of the FreeR reflections!

I suggest to replace the section header "Cell dimensions" with "Diffraction data" and to move the lines "R-meas (%)", "CC 1/2" and "Wilson B-factor (Å²)" up into this section, where they belong. The authors may want to replace the term "Redundancy" with the better term "Multiplicity". Finally, I note that the Ramachandran angles should add up to 100%.

Table S2 was revised according to the reviewer's suggestions and Nature Comm. format requirements.

Reviewer #3 (Remarks to the Author):

General comment.

This manuscript describes a very thorough and well-controlled set of experiments that clearly establish that cAMP is not the natural Trypanosome ligand for TbPKAR. It has however 2 shortcomings:

Novelty:

They clearly state:

"In bacterial transcription factors, some CNB domains can bind other ligands like heme in the case of

the CO sensing transcription activator CooA (4) or chlorinated phenolic compounds in CprK, a member of the ubiquitous CRP77 FNR family of transcription activators (5).”
So, the concept that CNB domains can in fact bind non-cAMP ligands has been established, albeit in bacteria.

We would like to emphasize that there is not a single report of eukaryotic CNB domains binding ligands other than cyclic nucleotides, and we also characterize the first PKA (of very many studied) not activated by cyclic nucleotides. We discuss this in the context of the few reports on distantly related prokaryotic CNB domains that have been shown to bind alternative ligands. We think that the proposal of more versatile use also of eukaryotic CNB domains for unidentified ligands and the biochemical properties of the first cAMP-independent PKA ortholog from *Kinetoplastida* are clearly novel. Novelty can be an arguable notion.

Discussion lines 540-541: «This challenges the current view that all PKA orthologs are cAMP-dependent » Surely that view was already challenged refs 4 & 5?

Unfortunately, this comment is due to a misunderstanding: refs 4 & 5 report CNB domain containing bacterial transcription factors, there is no mention of PKA, that anyway does not occur in prokaryotes. What we would like to challenge is the current view that PKA and cAMP-dependent protein kinase are synonymous protein names in all eukaryotic organisms.

What then is the natural ligand?

The shortcoming of not knowing what the real Trypanosome ligand activates PKA is reflected both in the title:

“Nucleoside analog activators of cyclic AMP-independent protein kinase A of Trypanosoma »

And in the Discussion:

Lines 664-665: “Is trypanosome PKA the effector of an unknown alternative second messenger? »

Lines 677-678: “The identity of the physiological PKA ligand and the respective pathway will be an exciting future research question.”

We fully agree that identity of the physiological PKA ligand is an important follow-up question, but see below

Line 330-334: “The evidence for a direct mode of action provided by in vitro kinase assays was further corroborated by measuring binding parameters between the activating compounds and purified N-terminally truncated *T. brucei* PKAR(199-499) expressed in *E. coli* (S4f Fig).”

It’s a shame then that they didn’t exploit binding to the CNB in TbPKAR to identify the natural in vivo ligand.

We actually attempted affinity purification of metabolite fractions from *T. brucei* on immobilized PKAR(199-499) expressed in *E. coli* as suggested above by the reviewer. Limitations of this approach are set by the amount of cell material that can be obtained from bloodstream *T. brucei*, the sensitivity of MS and the unknown concentration of the putative ligand. We were simply not successful. We would like to ask the reviewer to be generous and consider the following example: my lab has identified the quorum sensing activity of trypanosomes called “stumpy induction factor” (SIF) in 1997 (Vassella et al., J Cell Sci) and predicted some chemical properties from partial purification. It was only in 2018 that the Matthews group identified the putative ligand, mainly by genetic experiments (Rojas et al., Cell 176, 1–12). The history of SIF has been reviewed in a spotlight in Parasitology Today 35, p7 by Solleis and Marti in January 2019. We feel that the identification of an endogenous ligand is beyond the scope of this first report on a very unusual PKA.

Minor points:

Lines 137-138: “PKAC1 was Ty1-epitope tagged in situ and the second PKAC1 allele was deleted to generate *T. brucei* cell line $\Delta c1/Ty1-C1$ (Fig 1b). » I think they mean PKAC2?

We have changed to “One *PKAC1* allele was Ty1-epitope tagged *in situ* and the second *PKAC1* allele was deleted to generate *T. brucei* cell line $\Delta c1/Ty1-C1$ (Fig 1b)” (lines 130-131). This should make clearer that PKAC1 was tagged *in situ* in a hemizygous *PKAC1* knock out. The paralog PKAC2 was not touched.

Lines 489-491: “Early changes in cellular protein abundances elicited by PKA activation were identified by label-free quantitative proteomics. » What is the logic of looking at protein abundance as a readout of PKA activity?

As readout for PKA activity, we have now compared PKA phosphorylation targets (Fig. 5, new additions, and new Supplementary Fig. 8). The PKA-dependent proteome analysis was performed independently to indicate downstream processes preferentially activated or repressed by PKA in trypanosomes, as consequence of PKA phosphorylation activity. As expected, the large number of induced phosphorylations translates into a much smaller number of expression changes within the first 12 h that we analyzed.

Lines 512-513: "Therefore, 7-CN-7-C-Ino has higher PKA specificity with respect to target protein regulation » Now we've switched from abundance to regulation.

Sorry for the confusion, the argument has been rephrased (lines 426-433).

Accessions for referees

PRoteomics IDentifications (PRIDE) database: <https://www.ebi.ac.uk/pride/archive/>

Phosphoproteome dataset

Project Name: Label-free, quantitative analysis of PKA-dependent phosphoproteome changes in *Trypanosoma brucei*

Project accession: PXD012245

Reviewer account details:

Username: reviewer95996@ebi.ac.uk

Password: XZjX28Lg

Proteome dataset

Project Name: Label-free, quantitative analysis of PKA-dependent proteome changes in *Trypanosoma brucei*

Project accession: PXD009073

Reviewer account details:

Username: reviewer75041@ebi.ac.uk

Password: as4YD4iR

Reviewers' Comments:

Reviewer #1:

None

Reviewer #2:

Remarks to the Author:

- / -

Reviewer #3:

Remarks to the Author:

the authors have respond to my major scientific criticism of measuring protein abundance as a readout of PKA activity by doing phosphoproteomics following stimulation of PKA by their novel activating ligand.

Also, I accept their argument that describing a non-cAMP binding to the CNB domain in eukaryotes (albeit a lower eukaryote) is sufficiently novel and their argument that screen for the endogenous ligand us the CNB domains is beyond the scope of the present manuscript.

For me then, this revised manuscript is now acceptable for publication.